# Artificial Phantasia:
# Evidence for Propositional Reasoning-Based Mental Imagery in Large Language Models

## Abstract

This study offers a novel approach for studying complex cognitive behavior in artificial systems. Almost universally, Large Language Models (LLMs) perform best on tasks which may be included in their training data and can be accomplished solely using natural language, limiting our understanding of their emergent sophisticated cognitive capacities. In this work, we created dozens of novel items of a classic mental imagery task from cognitive psychology. The task consists of following a series of short instructions (3-5 steps), performing basic transformations on imagined letters and simple shapes to create a mental image of an object, and finally recognizing and labeling the object. Traditionally, cognitive psychologists have argued that this task is solvable exclusively via visual mental imagery (i.e., language alone would be insufficient). LLMs are perfect for testing this hypothesis. First, we tested several state-of-the-art LLMs by giving text-only models written instructions and asking them to report the resulting object after performing the transformations in the aforementioned task. Then, we created a baseline by testing 100 human subjects in exactly the same task. We found that the best LLMs performed significantly above average human performance (9.4%-18.2% increase over the human average of 54.7%, $p < .00001$). Finally, we tested reasoning models set to different levels of reasoning and found the strongest performance when models allocate greater amounts of reasoning tokens. These results provide evidence that the best LLMs may have the capability to complete imagery-dependent tasks despite the non-pictorial nature of their architectures. Our study not only demonstrates an emergent cognitive capacity in LLMs while performing a novel task, but it also provides the field with a new task that leaves lots of room for improvement in otherwise already highly capable models. Finally, our findings reignite the debate over the formats of representation of visual imagery in humans, suggesting that propositional reasoning (or at least non-imagistic reasoning) may be sufficient to complete tasks that were long-thought to be imagery-dependent.

## 1 Introduction

Large Language Models (LLMs) have progressed exponentially in the last few years. However, the most popular benchmarks rely on reading comprehension (Kočiský et al., 2018), information recall (Rein et al., 2023; Hendrycks et al., 2021), logical reasoning (Wang et al., 2024b), coding (Khan et al., 2023), or other similar tasks in principle solvable only through text and prone to data contamination risks from preexisting information in their training dataset (Sainz et al., 2023). In recent months, questions about the scalability of complexity, the appropriate usage of reasoning tokens, and the design and evaluation of reasoning tasks have been raised despite LLMs performance (Shojaee et al., 2025; Lawsen, 2025). Here, we present a mental imagery task adapted from cognitive psychology that offers an opportunity for assessing models in novel and sophisticated ways. We designed bespoke stimuli that we can be certain are not in their training data and that, as we show below, leave room for great improvement in models' performance (despite the best models outperforming humans). Moreover, our results suggest that LLMs may be capable of performing a mental imagery task long-thought to be unsolvable by relying solely on language, offering a new challenge for cognitive psychology.

## 1.1 Pictorial vs. Propositional Mental Imagery

Cognitive psychologists have been embattled in the last fifty years in a heated debate about the nature of mental imagery. Two opposing camps have proposed that mental imagery's format is either pictorial (e.g., Kosslyn (1973; 1996)) or propositional (e.g., Pylyshyn (1973; 2002)). In general, a propositional format would be one in which the representation is composed of discrete symbolic elements that can be combined to create new, complex representations (e.g., the word "red" and the word "house" are discrete symbols that, when combined, can create the complex representation "red house"). Advocates of the propositional view argue that the visual information contained in mental images can be captured by propositional descriptions (i.e., discursive elements) in a symbolic so-called "language of thought". In consequence, tasks used to prove mental images are in a pictorial format can all, after all, be solved through language or reasoning alone. Pictorialists disagree. They consider some representations are non-symbolic and non-discrete: analogue, continuous representations that do not have a canonical decomposition (e.g., "red house" is naturally parsed as "red" plus "house", but how are we to systematically parse the many strokes of the Mona Lisa?). Rather, parts of pictorial mental representations correspond to parts of what they represent and they represent multiple properties simultaneously (Quilty-Dunn, 2019).

One of the foundational studies of pictorial mental imagery is an object reconstruction task (Finke et al., 1989). Subjects were required to visualize in their mind's eye a series of combinations and transformations of basic shapes and letters. Subjects then indicated what object the resulting construction of shapes looked like in a free verbal report. The structure of this task is straightforward. Simple shapes and letters are given in stages to the subject where transformations of the existing figures (or the entire scene) follow (Figure 1). Between 2 and 4 sets of transformation instructions are provided before the subject is asked to identify the final image.

The pictorial view of mental imagery vastly dominates psychology today (Pearson & Kosslyn, 2015; Pearson et al., 2015; Block, 2023; Zeman, 2024). According to it, success in this task (and others like it) is only possible through the use of visual pictorial imagery (as opposed to logical or propositional reasoning, but see Pylyshyn (2002)). The pictorial view has gained dominance in the field through its appeal to evidence from neuroimaging studies showing similarities in neural activity between visual and imagery tasks (Pearson et al., 2015; Naselaris et al., 2015; Dijkstra et al., 2019) and, crucially, from mental imagery tasks (Shepard & Metzler, 1971; Kosslyn, 1973; Finke et al., 1989; Pearson & Kosslyn, 2015). For example, in the Finke et al. object reconstruction task, the final identification step is supposed to require a properly constructed *image* that subjects can simply read off from their mind's eye. Confidence in this view is so strong that some proponents go as far as to think that solving this kind of visualization tasks is "virtually impossible to do without using [pictorial] imagery" (Finke, 1990, p. 19).

| | Step 1 | Step 2 | Step 3 |
|---|---|---|---|
| **Instructions** | Imagine a capital letter "D". | From there, imagine the figure rotated 90 degrees to the left. | From there, imagine a capital letter "J" attached to the bottom center of the figure. |
| **Mental Image** | D | ◠ | ☂ |

Figure 1: One of the instruction sets introduced in Finke et al. (1989). Here, subjects are meant to recognize from the resulting mental image that the final imagined object looks like an *umbrella*. The instructions have been rewritten to be clearer both for prompting LLMs, as well as for human understandability.

## 1.2 Solving Mental Imagery Tasks without Mental Imagery?

Many tasks in everyday life involve the usage of mental imagery to some degree (e.g., navigation, planning and decision-making, mental simulation, episodic memory, organization, spatial reasoning, emotional engagement and regulation, among others; Bocchi et al. (2017); Shepard & Metzler (1971); Palombo et al. (2018); Wheeler et al. (2000); Byrne et al. (2007); Holmes & Mathews (2010); Krasich et al. (2024)). Unsurprisingly, most humans report having conscious mental imagery. However, a small percentage (1-4%) of the population—aphantasics—report no conscious mental imagery (Wright et al., 2024; Faw, 2009; Zeman et al., 2015; Dance et al., 2022; Zeman, 2024). If the pictorial view were correct, we would predict aphantasics to be incapable of performing mental imagery tasks at all; but this is not what we find (Blomkvist, 2023; Kay et al., 2024; Pounder et al., 2022; Bainbridge et al., 2021). Aphantasics perform (almost) at the same level as people with imagery. While there is a possibility that they rely on unconscious visual mental imagery (Nanay, 2021; Michel et al., 2025), aphantasics tend to report that they use verbal strategies (Keogh et al., 2021; Kay et al., 2024), giving renewed credence to the possibility of a purely propositional mental imagery.

In the human mind, language and imagery are deeply intertwined. It can be hard to evaluate the introspective reports of subjects (aphantasic or not) about the strategies that they use, as humans in general do not have good access to the inner processes that support their behavior (Nisbett & Wilson, 1977); or they may confabulate about the actual contents of their mental images (Bigelow et al., 2023). State-of-the-art artificial systems such as LLMs offer a unique opportunity to test a system whose architecture and processing is primarily propositional, specifically, linguistic. Nothing truly analogous to a visual (i.e., pictorial) mental imagery seems available to LLMs.[1] Does this mean that there are types of reasoning that are just not available to them? Or is it possible that LLMs rely solely on their language-trained and language-processing architecture to achieve similar goals as humans who experience mental images? After all, as mentioned earlier, aphantasics are reported to perform at the same levels as imagers in a plethora of tasks previously thought to require mental imagery. The imagery debate does not seem, after all, to have been completely settled.[2]

## 1.3 Motivation for LLM Mental Imagery Tasks

To test whether LLMs are capable of solving tasks designed to probe pictorial mental imagery despite relying exclusively on text processing, we gave several state-of-the-art models (Claude, Gemini, OpenAI), as well as several open-weight models (DeepSeek, Qwen, GPT oss), expanded, bespoke instruction sets following Finke et al. (1989)'s approach described above. We also asked models with image capabilities to generate images in each step and to consider them in their answers. As the object reconstruction task we used is compositional (different images or aspects of images from each step need to be combined in subsequent steps to obtain the final answer), we conjectured that image-aided reasoning could increase performance, especially if the pictorial imagery framework is correct, and if forcing the models to produce and consider images in the intermediate steps could alter their approach to the task. (See Appendix C for further discussion about this approach.) Finally, we obtained a human baseline for for this task by testing 100 human subjects.

Whether LLMs can perform at a human level on this task is of intrinsic interest to understand what these new models can achieve. This type of object reconstruction is an ideal challenge for LLMs. The task is structured entirely through natural language (both the input and the output); the results are easily evaluated; and we included newly created examples that could not possibly be in their training set. Moreover, due to the nature of the task and the fact that it can be expanded, a human baseline can be straightforwardly established at any point. Additionally, LLMs performing at or beyond the human baseline would constitute evidence for LLM propositional reasoning-based imagery. The

---

[1] While recent multimodal extensions have been trained not just on linguistic corpora but on images as well, and they can take images as input and produce them as output, they still rely on high-dimensional embeddings that are not visual in nature Kiela & Bottou, 2014; Kim et al., 2020; Radford et al., 2021. Current models certainly do not have a dedicated visual module built-in and they do not process tokens in a visual format.

[2] Aphantasia research has recently re-opened questions regarding the nature of mental imagery representations (Lorenzatti, 2025; Lebon, 2025). Proposals have ranged from unconscious pictorial representations (Michel et al., 2025; Nanay, 2021) and absent pictorial representations to a preserved spatial imagery despite a diminished or absent object imagery Bainbridge et al., 2021; Phillips, 2025. The door for non-pictorial mental imagery has certainly reopened.

results from this task are also of interest for the cognitive science debate about the format(s) of mental imagery, since it would put to a test the idea that mental imagery necessarily involves some pictorial component.

## 1.4 RELATED WORK

Recent publications exploring LLMs' capacities have included notable work on classic cognitive tasks, e.g., evaluating spatial cognition as an emergent property of frontier models (Ramakrishnan et al., 2025; Wu et al., 2024; Ivanova et al., 2025), or measuring the stability of psychometric attributes (Li et al., 2024). Additionally, benchmarks for spatial and visual reasoning have been established (Chollet et al., 2025). More broadly, emergent cognitive properties of artificial systems have also been studied beyond just those of the transformer architecture. There is a wealth of accumulating evidence showing that Deep Neural Networks (DNNs) are useful tools for understanding human cognition in general (Demszky et al., 2023; McGrath et al., 2024; Leshinskaya et al., 2025; Frank & Goodman, 2025) and visual perception in particular (Chen & Bonner, 2025; Yamins et al., 2014). Finally, recent attempts to enhance LLMs' visual reasoning capacity (e.g., spatial planning, visual completion) have explored the capabilities of visual-language models (Yang et al., 2025). While this last study offers an interesting approach, here we focus on the foundational question of whether mental imagery tasks (rather than visual tasks) can be solved by frontier LLMs on the sole basis of linguistic manipulation. We discuss several other related topics in Appendix C.

## 2 EXPERIMENTAL DESIGN

We used 60 instruction sets for an object recognition task. The set consisted of 48 completely novel examples, which we created specifically for this study, and the remaining 12 were taken from the original study by Finke et al.. Our new items varied in number of steps (2 to 4), final object, and overall complexity (Figure 2). We aimed to keep Finke et al.'s restriction that the final shape should not be identifiable until after the final transformation. Some of the tasks resulted in the same (or very similar) final shapes (e.g., three different ways of arriving at a shape that represented "glasses" or "binoculars"), but as all combinations were unique and involved novel transformations, we believe this is not of concern.

| | Step 1 | Step 2 | Step 3 | Step 4 |
|---|---|---|---|---|
| **Instructions** | Imagine two lowercase letter "o" next to each other. | From there, imagine the two figures pressed together so that they are just touching. | From there, imagine another lowercase letter "o" just touching the top of the figure in the center. | From there, imagine a lowercase letter "v" just touching the bottom of the figure in the center. |
| **Mental Image** | | | | |

Figure 2: One of our new instruction sets demonstrating the slightly increased cognitive complexity and more ambiguous canonical form ("balloons", "flower bouquet", or "ice cream", among others). Note the usage of two letters in the first step, the abstract reference to existing symbols and scenes, and the final shape not being determinable until the final step.

Our new items integrated several changes to the original ones developed by Finke et al. (1989). Most notably, we allowed one step to include up to two letters, rather than just one, thus increasing cognitive load (Miller, 1956; Farrington, 2011) but allowing more varied scenes. Additionally, we did not restrict the final image to having only one canonical form. The intended canonical form was not always immediately obvious, though at least one form was always clear. Our items' difficulty had a wider range, which we confirmed after establishing a new human baseline (see Appendix 2.3). The complete instruction-sets are included in the project's anonymous GitHub repository (see Appendix B).

Finally, we updated the language in the 12-items taken directly from Finke et al. to match the format of our improved versions. Notably, ambiguous language was clarified (e.g., specification of capital versus lowercase letters), subsequent instructions were modified to reference earlier instructions only abstractly (e.g., "the existing symbol" versus "the 'E'"), and 180 degree rotations were changed to flips (when vertical) or mirroring (when horizontal). We did not ask models or human subjects to guess the resulting shape at each step, unlike the original experiment, only its final form.

We should highlight that because these 48 new items were created *ex novo* for this experiment, it is highly improbable that any of them were present in the training data of any of the LLMs, and it is materially impossible that all of them were. This is of crucial importance for testing the emerging reasoning capacities of these types of models.

## 2.1 IMAGE-AIDED INSTRUCTIONS

For each model that had a compatible image generation pipeline (excluding GPT-5[3]), we ran a modified version (Image-Aided) of the standard version (Language-Only) described above. In this modified image-aided version, models were prompted to generate images and modify those images (see Supplemental Figure S1 for an example), rather than imagine. Whenever possible, we kept reasoning enabled within the models. For Gemini we used the native image generation capabilities of 2.0 Flash's image generation preview version, and, for OpenAI, we used the native GPT-image-1 image generation tool integration.

## 2.2 MODEL SELECTION

We gave the 60 instruction sets to three consumer accessible groups of models: Claude, Gemini, and OpenAI. For OpenAI reasoning models, 'reasoning' was enabled and 'reasoning_effort' set to 'high'. For older Gemini models, 'thinking' was set to 'dynamic'. For Gemini 3 Pro, 'thinkinglevel' was set to 'high'. For Claude Sonnet 4 we allocated 4000 tokens for extended thinking; for Claude Opus 4.1 we allocated 9000. For models that allow temperature modification, the value was set to 0.1 (we discuss some impacts of this in Appendix E.3). All other parameters were kept at default values. Our specific model choices are outlined in Supplemental Table S1.

For our initial analysis, we performed each experiment twice: once in a single context for all instruction sets (single-context), and once with a new context for each instruction set (multiple-context). This was done to see whether there was any significant difference in performance due to in-context learning (Dong et al., 2024; Wurgaft et al., 2025). The instruction sets were presented to each model in the same random order.

After completing our testing of the proprietary consumer models, we further tested four open-weight models: Qwen 3, Qwen 3 VL, DeepSeek R1, and gpt-oss-120b. We left all parameters to their default specifications, with exception of the 'reasoning_level' parameter for gpt-oss-120b which we set to 'high'.

## 2.3 HUMAN BASELINE

We recruited 100 adult participants online. Each was given a random subset of 15 of the 60 instruction sets (Finke et al. sets and our 48-Item expansion set) for a total of 1500 submitted answers. Each instruction set had between 21-27 responses. The instruction sets were administered through a Qualtrics XM survey in a random order. To prevent textual biases from impacting the results (e.g., seeing a 'd' on a screen affecting the specific shape of the imagined 'd'), the instructions were played through audio recordings. Each occurrence of a letter in an instruction was modified to include the corresponding phonetic word from the International Radiotelephony Spelling Alphabet to increase the clarity of individual instructions (e.g., "Imagine a B, as in Bravo."). All audio was recorded by the same speaker (Author 'Anonymous') and there was only one version of each audio instruction. If the same instruction appeared in two different sets, each set was given unique recordings. Participants were able to listen to the instructions as many times as needed, and were asked to imagine

---

[3]GPT-5 was released after we collected data on our image paradigm and found that it did not succeed for other models. As no further upgrades for GPT-image-1 were additionally released we chose not to test GPT-5 with GPT-image-1.

shapes without any visual aids (e.g., drawing). Finally, participants completed the Vividness of Visual Imagery Questionnaire (VVIQ) to assess their overall mental imagery capacity (Marks, 1973) (see Appendix D.3). Subjects participated for payment and were recruited through the online platform Prolific with an evenly distributed gender quota. To ensure intelligibility of the instructions, only subjects from the United States who reported English as their first language were sampled.

## 2.4 PERFORMANCE EVALUATION

Given the subjective nature of the task (e.g., Are "balloons", "flower bouquet", or "ice cream" all equally valid as an answer for the instructions in Figure 2?) and the potential for correct but variable answers, we recruited 376 naïve subjects on Prolific to grade answers (LLM and human) from the studies described above. Subjects were shown a label along with a corresponding image approximating the final outcome of a set of instructions. They were asked to rate the reasonableness of the label describing the image we created *in lieu* of target mental images. We collected 2030 unique answers from LLMs and humans in our mental imagery experiments. We excluded 122 of these answers because they were nonsensical, explicitly non-answers (e.g., "unsure", "I don't know"; see Appendix E.2 for discussion), restatements of the instructions, or sexually explicit. We ended with a final set of 1911 valid answers to use as labels. Each subject grader was given 30 of these labels with the prompt: "How well does this image represent <label>?" and they were asked to respond on with a 5-point scale: "Not at all", "A little", "Moderately", "A lot", "Completely". In addition, the authors provided independent "expert" evaluation of all of the labels. The final score used to grade each valid response in our study was a weighted average of the expert's grades and the outsourced grades given by the Prolific subjects (see Appendix D.1).

## 3 RESULTS

### 3.1 HUMAN PERFORMANCE

Humans subjects exhibited an average performance of 54.7% (of the maximum possible score) in our 60-item task. We established that all of the 48 new instruction sets were doable (though the hardest ones only received one or two meaningful answers from the entire subject population). Subjects received close to perfect scores on the easiest instruction sets, with only one or two non-meaningful answers. This offered us a good range of difficulties which was optimal for testing both human and artificial model abilities. (See Appendix D.2 and Supplemental Figure S11 for more information on item difficulty estimation and difficulty distribution).

When we looked exclusively at our 48 newly designed items, we observed a comparable performance between our subjects' performance and that of Finke et al.'s subjects in their original 12 items (52.6% and 56.1% of the maximum possible score, respectively).[4]. Notably, our subjects' performance on the 12 items following Finke et al.'s was higher (63%). This difference could potentially owe to the fact that our set has a wider range of difficulty levels both in terms of number of steps per trial and in terms of compositionality, complexity of transformations, and recognizability of the final shape (all of which were explicitly desired traits in our experiment), or it could be simply due to our rewritten instructions.

Finally, subjects' VVIQ scores were well within expected bounds and we found a small negative correlation between performance and mental imagery capacity VVIQ (see Appendix D.3 for further discussion of VVIQ results).

### 3.2 RAPID ADVANCEMENT IN 2025

OpenAI's o3 model family, as well as GPT-5, vastly outperformed every preexisting model, surpassing the human baseline (humans were 9.4% below o3, which performed at 64.1%, $\chi^2 = 28.631$, $p < .00001$, $CI = [-0.128, -0.06]$; humans vs o3-Pro: $-11.9\%$ difference, $\chi^2 = 45.76$, $p < .00001$, $CI = [-0.153, -0.086]$; humans vs GPT-5: $-12.3\%$ difference, making GPT-5 the

---

[4]In Finke et al., each participant provided responses for six items. We normalized Finke et al.'s scores to our scoring-system by assigning 5 points for each correct and 1 point for each incorrect answer to each label (the minimum score in our paradigm)

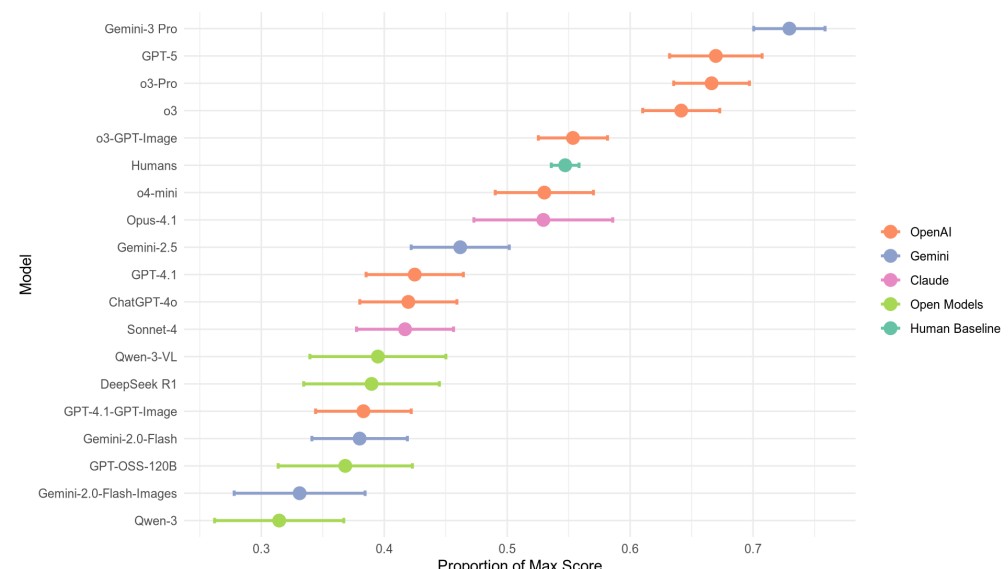

Figure 3: Performance results in humans and LLMs. Data shows proportion of maximum possible score for all tested models. Only Gemini 3 Pro, GPT-5, and the o3 family significantly surpass the human baseline. Error bars indicate 95% confidence intervals.

best model at 67%, $\chi^2 = 33.302$, $p < .00001$, $CI = [-0.163, -0.082]$). Furthermore, Gemini 3 Pro reached a new frontier on our task: demolishing the human baseline by 18.2% (humans vs. Gemini 3 Pro: $-18.2\%$ difference, $\chi^2 = 108.103$, $p < .00001$, $CI = [-0.214, -0.151]$).[5] We show the stark gap between o3, GPT-5, Gemini 3 Pro, the human baseline, and every other model we tested in Figure 3. Based on our scoring methodology (Appendix D.1), we graded the outputs of each model (and the human baseline) and calculated the maximum possible score (see Table 1 for full scores; see Supplemental Table S5 for results of the 48-novel item subset and Supplemental Table S4 for results of the 12-items following Finke et al.).

The only other models to perform non-significantly different from the human baseline on our task were o4-mini ($\chi^2 = 0.577$, $p = .4477$, $CI = [-0.025, 0.059]$) and Claude Opus 4.1 ($\chi^2 = 0.299$, $p = .5845$, $CI = [-0.042, 0.077]$). Somewhat surprisingly, we found that adding images to o3 significantly decreased the model's performance although it remained comparable to the human baseline (o3 vs. o3 with GPT-image-1: 8.8% difference, $\chi^2 = 16.139$, $p < .0001$, $CI = [0.045, 0.131]$, humans vs o3 with GPT-image-1: $\chi^2 = 0.143$, $p = .7057$, $CI = [-0.037, 0.024]$). The older or cheaper models, however, performed very poorly across the board. We looked at two other models from OpenAI: ChatGPT-4o and GPT-4.1 (with and without images), both performed significantly worse than the o3 model family (see Appendix E.4.1). Claude Sonnet, Gemini 2.5 Pro, and Gemini 2.0 Flash all performed poorly. Gemini 2.0 Flash with images performed the worst of all proprietary models we tested.

Our testing of open models was extremely disappointing with all models performing worse than most, if not all, of the proprietary models. We discuss potential reasons for why this may be the case in Section 3.5.

Finally, we also measured the effect of the 'reasoning_effort' parameter on the best performing OpenAI models. We found that higher reasoning effort led to improved results on our task (see Appendix E.1 and Supplemental Figures S4 and S5).

---

[5]Statistical significance in the main results presented in Figure 3 and Table 1 was determined after correcting for multiple comparisons using Bonferroni: alpha = 0.05 / 18 = 0.0028.

Table 1: Human and LLMs scores

| Agent | Score | Max Possible Score | Proportion |
|---|---|---|---|
| Humans | 4098.217 | 7490 | 0.5472 |
| o3*** | 577.390 | 900 | 0.6415 |
| o3 + GPT-image-1[ns] | 664.182 | 1200 | 0.5535 |
| o3-Pro*** | 599.621 | 900 | 0.6662 |
| GPT-4.1 | 254.862 | 600 | 0.4248 |
| GPT-4.1 + GPT-image-1 | 229.827 | 600 | 0.3830 |
| o4-mini[ns] | 318.131 | 600 | 0.5302 |
| Gemini 2.5 Pro | 227.074 | 600 | 0.4618 |
| Gemini 2.0 Flash | 227.982 | 600 | 0.3425 |
| Gemini 2.0 Flash + Images | 99.345 | 300 | 0.3800 |
| Claude Sonnet 4 | 250.140 | 600 | 0.4169 |
| Claude Opus 4.1[ns] | 158.819 | 300 | 0.5294 |
| GPT-5*** | 401.883 | 600 | 0.6697 |
| **Gemini 3 Pro*** | **656.589** | **900** | **0.7295** |
| DeepSeek R1 | 116.900 | 300 | 0.3897 |
| Qwen 3 | 94.358 | 300 | 0.3145 |
| Qwen 3 VL | 118.442 | 300 | 0.3948 |
| gpt-oss-120b | 110.483 | 300 | 0.3682 |

Model in bold indicates highest performer in the group; the human baseline is in green; models in purple surpass the human baseline significantly [*** $p < .001$]; models in blue are not significantly [ns] different from the human baseline and the rest in white are significantly lower [$p < .001$]. Significance and non-significance determined after Bonferroni correction for multiple comparisons. See Appendix E.4.1

## 3.3 IMAGE-AIDED REASONING

Generating images, and solving the task using them, produced disappointing results. Perhaps unsurprisingly, in all cases of our image-aided paradigm the model performance dropped or, at best, stayed the same: o3 vs. o3 with GPT-image-1 (8.8% difference; $\chi^2 = 16.139$, $p < .0001$, $CI = [0.045, 0.131]$); GPT-4.1 vs. GPT-4.1 with GPT-image-1 (4.2% difference; $\chi^2 = 1.999$, $p = .1574$, $CI = [-0.015, 0.099]$); Gemini 2.0 Flash vs. Gemini 2.0 Flash with images (4.9% difference; $\chi^2 = 1.854$, $p = .1733$, $CI = [-0.02, 0.117]$). We note, however, that o3's strong performance allowed greater room for a significant decrease. It is unclear what the overall effect of image-aided reasoning is, as the models still found some success (though diminished), and more exploration of its effects is needed (Yang et al., 2025; Wu et al., 2024). Notably, modification of the 'reasoning_effort' hyperparameter led to almost identical results, unlike what happened with the standard models without images (See Appendix E.1.)

## 3.4 MULTIPLE- VS. SINGLE-CONTEXT

We initially ran each model (excluding Gemini 2.0 Flash with images and o3 with images due to token limitations) in both a single context for all instructions as well as in a new context for each set of instructions (multiple-context). In our analysis of o3 and o3-Pro we found that including previous examples in context did not significantly change the overall performance (SC vs. MC in o3 $\chi^2 = 0.1064$, $p = .7443$, $CI = [-0.056, 0.083]$; and in o3-Pro $\chi^2 = 0.02$, $p = .887$, $CI = [-0.061, 0.075]$). This indicates that in-context learning was not significantly beneficial for this task. Because of this, we did not differentiate between context types for statistical analysis and instead analyzed results from both types of context together. Additionally, when we further tested several additional models with different reasoning levels, we only tested them in multiple-context.

We graph the performance of the context interval variations of all tested models in Supplemental Figure S2.

### 3.5 OPEN MODEL FAILURE

Across the board, the open models we tested performed horribly. Qwen 3 was the single worst performing model, and no open model met or surpassed the human baseline. We are not entirely sure as to the cause of this (due to not having access to architectural specifications of proprietary models), but, nevertheless, we were able to learn at least one somewhat interesting point.

Qwen 3 and Qwen 3 VL did not have a significant difference in performance (after accounting for multiple-comparisons). Qwen 3 vs. Qwen 3 VL: $-8.03\%$ difference, $\chi^2 = 3.88$, $p = .0489$, $CI = [-0.16, -0.001]$. We found this notable due to the architecture of the model Bai et al. (2025); most critically, the vision-language alignment step. By aligning visual information to a text embedding, the model learns how to represent visual information textually (and therefore propositionally). The other open models did not include this step, and therefore, may have had reduced performance.

## 4 DISCUSSION

Our results were surprising. LLMs successfully accomplished a task in which humans are believed to rely (almost) exclusively on visual mental imagery. They also offer an interesting window into the capacities of advanced artificial language systems to provide insight on tasks that are thought to require something beyond what language models are capable of. Gemini Pro 3, GPT-5 and the o3 Language Reasoning Model family outperformed humans across the board. These models do not have any (known, immediate) access to any form of image processing built-in by default when dealing with text decoding. Additionally, we can infer that no image generation was being used unless explicitly called upon due to the decreased cost of simple inference. Thus, these models should not have been able to complete our task so successfully given the dominant pictorial views on mental imagery. It would seem, however, that these LLMs completed our task via language token manipulations.

However, there is an alternative explanation, potentially orthogonal to the pictorial and propositional accounts. On this view, mental imagery is not a monolithic capacity; rather it must be distinguished into two capacities: spatial and object imagery (Phillips, 2025; Teng, 2025). According to this distinction, imagining objects and their surface features is supported by neural and psychological mechanisms that are different from the mechanisms supporting imagining the *spatial relations* between objects. Evidence for this position comes from neurological patients with lesions resulting in preserved spatial reasoning abilities despite losing conscious mental imagery (Farah, 1988). It is also supported by behavioral work with aphantasics who exhibit normal results in spatial imagery questionnaires and (almost) normal performance in mental imagery tasks that rely heavily on spatial relations such as remembering a scene and mental rotation, which is clearly related to the task we used here (Bainbridge et al., 2021; Dawes et al., 2022). (See Appendix D.3 for discussion of aphantasic subjects in our sample.)

While LLMs' architecture, training, and data processing is primarily based on text, text-only models can acquire representations of spatial concepts (Patel & Pavlick, 2022) and develop internal representations matching a spatial layout solely through procedural descriptions of moving through spaces. Moreover, embeddings from different modalities (e.g., vision and text) can be mapped to a shared representational space (Radford et al., 2021; Girdhar et al., 2023). It is thus possible that while relying solely on textual token manipulation, the most advanced LLMs are still able to extract the required spatial relations of the objects in each step of our instruction sets to perform at or above human level. This interpretation may help bridge different theories of mental imagery.

Notably, there is some probability of the original 12 items in the Finke et al. task being in the training data. This, however, cannot easily explain away our results. First, only 12 out of 60 instruction sets appeared in their original paper; the rest of them are completely novel designs. Second, the novel items include rather idiosyncratic examples that would be astounding if anything like them were explicitly in the training data (e.g., a computer mouse made with an S, a C, a horizontal line, and a D). Third, and related to the previous point, even if the *type* of task is present in the training data, and even if LLMs have information about the shape of letters (e.g., they know an 'o' is like

a circle and they know ice cream can be represented with circles), the models still have to solve a challenging compositional task: They need to put together all the elements, in the right spatial layout, in some cases after 4 transformations including rotation and mirroring, and only then use the resulting composite novel shape to "read out" the actual answer. This demanding task goes well beyond knowing that ice cream icons have circles (notoriously, an enormous amount of other things can also be drawn using circles). Fourth, no model, not even the best performing ones, universally solved all the original Finke items, suggesting that the models are not simply finding a ready-made solution from their training data. Finally, in the case of some models (e.g., GPT 4.1 and o3) we have reasons to believe that they may share their training data (both have a June 01, 2024 knowledge cutoff). Even if some information relevant to our task were buried in the models' training data, it is the parameters and general architecture of the models that ultimately determine how well they were able to find relevant information and solve our novel items. Thus, our data shows that some advanced LLMs can reason about composite shapes and their spatial layout by relying solely on linguistic embeddings in ways that seriously challenge the hypothesis that a pictorial format is necessary for completing our task.

Frontier LLMs give a new perspective to the debate on the formats of mental imagery, re-opening questions about the necessity of iconic mental imagery and the adequacy of classic tasks to test it. Furthermore, our results also open an important question in computer science about the kind of tasks that our most advanced language models can accomplish providing an opportunity to create new, complex benchmarks in sophisticated cognitive tasks.

## 5 CONCLUSIONS

Advancements in Large Reasoning Models have progressed so quickly that testing all their possible emergent properties poses serious challenges. Our research shows how to test one such property: propositional reasoning-based mental imagery. We determined that Gemini Pro 3, GPT-5 and the o3 family of models is more than capable of solving problems traditionally thought to require visual imagery. Furthermore, we note the difference in capabilities between OpenAI, Claude, and Gemini models in this form of advanced reasoning. Our instruction sets were created *ex novo*, so we can be confident that the performance we recorded is not an artifact of contaminated training data. Additionally, models did not universally succeed on the original items, which *could* exist within their training data. Our results could be deeply impactful, both for the artificial intelligence community, for examining and identifying a capability not yet measured within the most advanced models, and also for the cognitive science community, for discovering results that provide insight into the strategies and techniques used to accomplish imagery-dependent tasks. Through further analysis and experimentation on Large Language Reasoning Models, we may discover more about different ways in which the human mind can work (see Appendix F for Future Work).

## 6 LIMITATIONS

Due to the bespoke nature of our task, the amount of data collected is insufficient to make strong conclusions about similarly performing models (e.g., to differentiate between the performance of o3-Pro and GPT-5). Additionally, as o3 and GPT-image-1 are very compute-heavy models, data collection takes significant time and computational resources, which is an important consideration both from a practical and a theoretical standpoint when benchmarking. These characteristics of the models and of our task prevent significant iterations of data collection until models improve enough for this to be feasible. Notably, GPT-5 reaches similar performance to o3-Pro in significantly shorter time, though with more reasoning tokens, pointing to GPT-5's higher processing efficiency. Similarly, even further advancement occurred with Gemini 3 Pro (faster and cheaper processing), despite it having the highest performance of all models. Finally, the state-of-the-art frontier models are all closed weight and closed architecture. We therefore cannot speculate about the underlying causes of our results or the difference in performance across models.

## 7 ETHICS STATEMENT

The experiments involving human participants were approved by the authors' Institutional Review Board. All data provided by the authors' that had any identifiable information has been removed. LLM usage related to this project is addressed in Appendix A.

## 8 REPRODUCIBILITY STATEMENT

In order to reproduce our results, we provide our experiment code and all data used in the production of this manuscript in Appendix B. We provide information on model selection and features in Table S1 and the exact versions of the models we used in Table S2. Furthermore, we provide information on the technical setup of our project in Appendix G, including information on our Python and R environments, exact model runs, hardware, and financial costs.

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

## A  Usage of Large Language Models

Outside of the usage as described in the rest of this paper, LLMs were used to generate code and documentation for running the experiments and statistical analyses. All code was manually checked and verified post-generation to ensure soundness.

In order to ensure prevention of data contamination: other than in testing, no LLM was given access to the instructions or any other data that may allow improved performance on the task. Additionally, we disabled sharing input and output data with the parent company of each model we used in our experiment whenever possible.

## B  Access to Code and Data

Our code and data is accessible under MIT License in our anonymous GitHub repository (https://anonymous.4open.science/r/artificial_phantasia-825C/). All human subject data has been de-identified. Please refer to the README.md file in the repository for usage and explanatory information regarding the repository.

We provide code to run the experiment, view and re-perform our data analysis, and generate surveys for Qualtrics XM. Our dataset includes the instructions used for our task (along with the intended canonical form), the de-identified data from both humans and LLMs, the de-identified response ranking data, and the image representations of each intended image (the images from Finke et al., 1989 are their original versions for replicability).

Furthermore we have included the reasoning outputs for each of the open weight reasoning models. These are provided for transparency, despite no analyses being performed upon them (due to the poor quality of responses from the open models).

## C  Extended Related Work

Since the advent of LLMs, many groups have explored their compositional components. Notably, this has included investigating the ability to answer compositional subproblems without successfully answering the overall problem (Press et al., 2023), evaluating linguistic compositional generation (Kim & Linzen, 2020), and evaluating compositional generation in images (Liu et al., 2022). No existing work has looked at the composition of image-like propositions in LLMs. Furthermore, research on how LLMs think in general (Shiffrin & Mitchell, 2023; Liu et al., 2025; Lombrozo, 2024; Pavlick, 2023) and how they may access their own processes Plunkett et al. (2025); Betley et al. (2025) is a crucial stepping stone for further understanding how LLMs may perform imagery tasks via propositional reasoning.

## D  Extended Performance Evaluation - Humans

### D.1  Grade Weighting

In our initial work on grading the task we ran into two key issues: first, the issue of subjectivity as described previously; second, some answers were extremely literal and, as such, did not represent a new construction. Our initial plan was to grade similarly to Finke et al. (1989), but the subjectivity issue necessitated us exploring different options.

After recruiting our first set of subjects, we discovered a key issue with the responses. Namely, hyper-literal responses (e.g., 'bd' after asking subjects to imagine a 'b' next to a 'd' and pressing them together) were graded very highly. We attempted to control for this by instructing subjects to rate such examples poorly, but, unfortunately, these were routinely graded highly. Our solution to this issue was to have the authors grade all 1911 responses in addition to the 376 naäve subjects.

Almost all responses had 5 subject ratings, a few responses had 6, a very small number 7, and a single response 4 due to the random distribution methods of Qualtrics XM. In all cases we generated a 'normal_score' from the mean of all of the responses. In addition, we generated an 'expert_score' from the mean of the authors' ratings. Our final 'overall_score' was the average of the 'normal_score'

and the 'expert_score'. All scores for all responses used in grading are available in our dataset (see Appendix B).

The subject ratings were distributed (Supplemental Figure S8) very heavily towards "Not at all" (or a grade of 1) due to most examples being difficult to justify (due to misconstructions, possible confusion in the case of humans, or possible hallucinations in the case of LLMs). The expert scores (Supplemental Figure S9) were distributed similarly, but with a noticeably higher occurrence of "Completely" (or a grade of 5). Knowledge of the creation of the items and their intended canonical result, along with a reduced bias towards the drawn representations of the intended mental image likely account for this.

## D.2 DIFFICULTY ANALYSIS

Concurrently with the experimental task, we asked all participants to report how clear they found the instructions, as well as how identifiable they found the final mental image (both on a 5 point scale). These ratings as well as their standard deviations as a measure of consistency, along with the ratio of unique responses given to each answer, the number of instruction steps, and the number of imagined objects, were used to rank each of the instruction sets on the following weighted difficulty scale:

$$
\begin{aligned}
\text{Item Difficulty} = \ &(0.20 \times \text{Total Instruction Steps}) \\
&+ (0.20 \times \text{Total Objects}) \\
&+ [0.15 \times (6 - \text{Mean Clarity Ratings})] \\
&+ [0.15 \times (6 - \text{Mean Identifiability Ratings})] \\
&+ (0.10 \times \text{Clarity Ratings Standard Deviation}) \\
&+ (0.10 \times \text{Identifiability Ratings Standard Deviation}) \\
&+ (0.10 \times \text{Unique Response Ratio})
\end{aligned}
$$

$$
\text{Unique Response Ratio} = \frac{\text{Unique Responses Per Label}}{\text{Total Responses Per Label}}
$$

Overall, we found that our set of instructions had a wide range of difficulty levels (see Supplemental Figure S11 for the distribution). We confirmed that the items that we designed to be harder (e.g., constructing a "computer mouse") indeed came out ranked as the most difficult, whereas the easiest ones (e.g., constructing a "ladder") were ranked as the easiest.

We can validate this by viewing the mean score given to each instruction set by the graders, and seeing a broad distribution (Supplemental Figure S10).

## D.3 VVIQ ANALYSIS

The VVIQ scale range is 16 (minimum) to 80 (maximum). Subjects had a mean VVIQ score of 55.8, confirming that our sample was normal given recent large sampling efforts (Wright et al., 2024). We did not find any correlation between VVIQ scores and subjects' performance (Pearson's $r = -0.17$, $p = .0942]$). If anything, there was a negative trend, suggesting that imagery capacity as measured by the VVIQ could not predict performance in a task designed to measure mental imagery. See Supplemental Figure S12.

Of the 100 subjects we tested, one subject qualified as a true aphantasic (VVIQ score of 16, the minimum). This subject, however, had the 5th highest score of all human subjects. While this constitutes a single data point (all other subjects had higher VVIQ scores), it raises the question of how this subject was able to complete the task. Recent data has shown that aphantasics can complete imagery tasks surprisingly well (Blomkvist, 2023; Kay et al., 2024; Pounder et al., 2022), even if the exact way in which they accomplish this or whether they use a single strategy across tasks is still unclear.

There were three subjects who qualified as low-imagers (or weak aphantasics) scoring between 17 and 32 on the VVIQ (23, 29, 29). Two of the subjects performed around the median score for humans (53rd and 56th), while the third was the 15th highest scoring individual in our dataset.

Lastly, we had seven subjects who had very high imagery (or hyperphantasics) scoring between 75 and 80 (the highest) on the VVIQ (one scored 77 the rest 80). These subjects had very mixed performance with three subjects performing above the median (18th, 20th, and 24th), and four subjects performing well below the median, and in a couple cases getting close to the bottom performance (77th, 81st, 91st, and 94th).

The high performance exhibited by the aphantasic subjects and the most advanced LLMs, in addition to the lack of correlation between imagery capacity (measured via VVIQ) and performance in the task, offers further evidence that mental imagery may operate in a propositional format. At the very least, this supposedly gold standard for probing pictorial imagining may not be as well suited for the task as generations of cognitive psychologists have thought. Naturally, much more data and analysis is needed to cement this conclusion. The current experiments with LLMs, and future ones as well, may help further our understanding of how aphantasics are able to accomplish these and other tasks. (For one possible explanation appealing to spatial imagery, see section 4 in the main text).

# E    EXTENDED PERFORMANCE EVALUATION - LLMS

## E.1    REASONING VARIATION

In addition to the selection of models we tested with 'reasoning_effort' set to 'high', we tested the best performing reasoning models on 'low' and 'medium' (and GPT-5 on 'minimal'), as well as two other models on 'medium' (see Supplemental Table S3).

Overall we saw that as reasoning token and time allocations decreased, so did the performance (o3 'high' vs. 'medium': 7.7% difference [$\chi^2 = 5.401$, $p = .0201$, $CI = [0.011, 0.144]$]; vs. 'low' 10.2% difference [$\chi^2 = 9.513$, $p = .0020$, $CI = [0.035, 0.169]$]; GPT-5 'high' vs. 'medium': 7.5% difference [$\chi^2 = 4.594$, $p = 0.0320$, $CI = [0.005, 0.145]$]; vs. 'low': 16.4% difference [$\chi^2 = 22.084$, $p < .00001$, $CI = [0.094, 0.235]$]; vs. 'minimal': 26.2% difference [$\chi^2 = 55.522$, $p < .00001$, $CI = [0.193, 0.332]$]; additionally see Supplemental Figure S4 and S5). One interesting data point, however, was the lack of any significant difference between 'high' and 'medium' in o3 with GPT-image-1 ('high' vs. 'medium': 0.5% difference [$\chi^2 = 0.008$, $p = .9281$, $CI = [-0.06, 0.07]$). This may hint at the possibility that the image paradigm has, despite models overall lower performance, the potential to provide some support solving this task. The issues with compositional generation inherent to image generation models, however, may hinder it overall (Huang et al., 2023). We provide statistics between humans and our reasoning model variations in Appendix E.4.2.

## E.2    REASONING UNDER UNCERTAINTY

One remarkable result we found was the lack of capability for LLMs to answer with uncertainty (and yet, it was unsurprising given well-known limitations of LLMs). We noted 54 occurrences of humans responding some form of "I don't know" to an instruction set, and 0 occurrences in the LLM responses. It is highly possible that LLMs provide responses even when great uncertainty exists (especially in our hardest instruction sets). This is an area worth exploring in further analyses on how LLMs succeed or fail at the task (Brahman et al., 2024; Kadavath et al., 2022; Malinin & Gales, 2021).

Interestingly, when asked to report the imagined object when given no instructions, open models reported answers like "blank slate" or "artificial neuron" implying some ability to give ambiguous responses in cases of uncertainty. The impact of this on our task needs to be explored further by future researchers.

## E.3    MODEL TEMPERATURE

In our initial testing, with the exception of Gemini 2.5 Pro (where we set the temperature to 0.1), all reasoning models were restricted to high temperatures by their respective APIs (1.0 for Claude

and OpenAI models). After considering that the performance of Gemini 2.5 Pro (well below the other frontier reasoning models) may be due to the low temperature, we performed a new test using Gemini 3 Pro.

Temperature has been shown to meaningfully affect reasoning outputs (Wang et al., 2024a), though on many tasks it does not necessarily improve the results (Renze & Guven, 2024). We performed a hyperparameter search of the 'temperature' parameter using values 0.1, 0.55, and 1.0 upon the release of Gemini 3 Pro. We were curious if modification in temperature would allow the model to explore more creative paths to the correct answer and succeed more often (Turing, 1950). We found no significant difference between any of the temperature values tested. See Appendix E.5.

### E.4 EXTENDED STATISTICAL ANALYSIS

#### E.4.1 STANDARD PARADIGM

Humans vs. ... :

- o3: $-9.4\%$ difference, $\chi^2 = 28.631$, $p < .00001$, $CI = [-0.128, -0.06]$
- o3 with GPT-image-1: -0.6% difference, $\chi^2 = 0.143$, $p = .7057$, $CI = [-0.037, 0.024]$
- o3-Pro: $-11.9\%$ difference, $\chi^2 = 45.76$, $p < .00001$, $CI = [-0.153, -0.086]$
- GPT-4.1: 12.2% difference, $\chi^2 = 32.987$, $p < .00001$, $CI = [0.08, 0.164]$
- GPT-4.1 with GPT-image-1: 16.4% difference, $\chi^2 = 59.482$, $p < .00001$, $CI = [0.123, 0.206]$
- ChatGPT-4o: 12.8% difference, $\chi^2 = 35.86$, $p < .00001$, $CI = [0.086, 0.17]$
- o4-mini: 1.7% difference, $\chi^2 = 0.577$, $p = .4477$, $CI = [-0.025, 0.059]$
- Gemini 2.5 Pro: 12.2% difference, $\chi^2 = 32.987$, $p < .00001$, $CI = [0.08, 0.164]$
- Gemini 2.0 Flash: 16.7% difference, $\chi^2 = 61.741$, $p < .00001$, $CI = [0.126, 0.209]$
- Gemini 2.0 Flash with Images: 21.6% difference, $\chi^2 = 53.296$, $p < .00001$, $CI = [0.16, 0.272]$
- Claude Sonnet 4: 13% difference, $\chi^2 = 37.392$, $p < .00001$, $CI = [0.088, 0.172]$
- Claude Opus 4.1: 1.8% difference, $\chi^2 = 0.299$, $p = .5845$, $CI = [-0.042, 0.077]$
- GPT-5: $-12.3\%$ difference, $\chi^2 = 33.302$, $p < .00001$, $CI = [-0.163, -0.082]$
- Gemini 3 Pro: $-18.24\%$ difference, $\chi^2 = 108.103$, $p < .00001$, $CI = [-0.214, -0.151]$
- DeepSeek R1: 15.7% difference, $\chi^2 = 28.182$, $p < .00001$, $CI = [0.099, 0.216]$
- gpt-oss-120b: 17.9% difference, $\chi^2 = 36.444$, $p < 0.00001$, $CI = 0.121, 0.236]$
- Qwen 3: 23.3% difference, $\chi^2 = 61.874$, $p < .00001$ $CI = [0.177, 0.288]$
- Qwen 3 VL: 15.2% difference, $\chi^2 = 26.355$, $p < .00001$, $CI = [0.094, 0.211]$

#### E.4.2 REASONING VARIATIONS

Humans vs. ... :

- o3 'high': $-9.4\%$ difference, $\chi^2 = 28.631$, $p < .00001$, $CI = [-0.128, -0.06]$
- o3 'medium': $-1.7\%$ difference, $\chi^2 = 0.273$, $p = .6014$, $CI = [-0.076, 0.042]$
- o3 'low': 0.8% difference, $\chi^2 = 0.273$, $p = .8349$, $CI = [-0.051, 0.067]$
- o3 with GPT-image-1 'high': -0.6% difference, $\chi^2 = 0.143$, $p = .7057$, $CI = [-0.037, 0.024]$
- o3 with GPT-image-1 'medium': -0.1% difference, $\chi^2 = 0$, $p = 1$, $CI = [-0.06, 0.057]$
- o3-Pro 'high': $-11.9\%$ difference, $\chi^2 = 45.76$, $p < .00001$, $CI = [-0.153, -0.086]$
- o4-mini 'high': 1.7% difference, $\chi^2 = 0.577$, $p = .4477$, $CI = [-0.025, 0.059]$
- o4-mini 'medium': 5.8% difference, $\chi^2 = 7.209$, $p = .0073$, $CI = [0.015, 0.1]$

- GPT-5 'high': -12.3% difference, $\chi^2 = 33.302$, $p < .00001$, $CI = [-0.163, -0.082]$

- GPT-5 'medium': $-4.8\%$ difference, $\chi^2 = 2.441$, $p = .1182$, $CI = [-0.106, 0.011]$

- GPT-5 'low': 4.2% difference, $\chi^2 = 1.854$, $p = .1733$, $CI = [-0.018, 0.101]$

- GPT-5 'minimal': 14% difference, $\chi^2 = 22.151$, $p < .00001$, $CI = [0.081, 0.198]$

## E.5 TEMPERATURE VARIATIONS

Gemini 3 Pro:

- Temperature of 0.1 vs. Temperature of 0.55: 2.6% difference, $\chi^2 = 0.379$, $p = .538$, $CI = [-0.049, 0.101]$

- Temperature of 0.1 vs. Temperature of 1.0: $-0.14\%$ difference, $\chi^2 = 0.097$, $p = .7556$, $CI = [-0.088, 0.059]$

- Temperature of 0.55 vs. Temperature of 1.0: $-4\%$ difference, $\chi^2 = 1.038$, $p = .3082$, $CI = [-0.115, 0.034]$

## F FUTURE WORK

This work has several future steps given its impact both within cognitive science as well as artificial intelligence.

First, in humans, aphantasics lack voluntary visual mental imagery, but report little-to-no issues in other areas of everyday life; often times living without knowing of the condition at all (Larner et al., 2024). We suggest a follow-up study on the strategies and techniques used by aphantasics in comparison to those used by LLMs. The lack of visual imagery parallels the inability for visual imagery inherent to LLMs, and by studying both groups, it may be possible to learn more about the need for pictorial imagery or the lack there-of to perform compositional tasks like the one we tested.

Second, as multimodal auditory, vision, and language models continue to advance, the capabilities of these models surely will as well. We propose continuing to evaluate the performance of the leading frontier models on our paradigm. Currently, given the bespoke nature of our stimuli, our task is not contained within the training data of any model as of now and, as such, is not subject to data contamination issues. However, as time passes this risk continues to increase. Creating more items in this kind of task requires certain ingenuity but they are sufficiently straightforward that new ones can be devised *ad hoc*. Further model evaluation should include tasks hidden from public datasets to truly measure advancement in propositional reasoning.

Third, humans provided a great limitation to the creation of trials that would serve for further exploring LLMs capacities. The general rule is 7 plus or minus 2 (or even less) objects can be worked with at once in human working memory (Miller, 1956; Farrington, 2011). LLMs are only limited by their contexts, which can hold vastly more information than any human. Without this limitation, it is unknown the extent of LLMs propositional reasoning capacity. As such, trials with many more than 3-5 steps and many more than 4 objects may be created, and, furthermore, may be necessary for future evaluation given GPT-5 and the o3 family's performance on the existing items. Conversely, we do not know to what extent our task necessitates working memory. It may be entirely possible for humans to work through an arbitrary number of iterative instructions, with an arbitrary number of components (overall, not per-step), given that each step creates a new imaginary scene (and therefore humans may only have to keep the previously imagined scene, and any new information, in mind).

Finally, in the future, as models progressively get stronger, open weight models that perform well enough on this task to compete with the GPT-5 and o3 family may exist. If this happens, we propose mechanistic approaches that look inside the models and directly examine the formats of representation. Can we find direct evidence of propositional reasoning (or learned iconic representations or a distinct spatial imagery capacity) by examining the weights of the models McCoy et al. (2019); Piantadosi et al. (2024)? Unfortunately, our results with some of the largest current open models, this possibility is not available since open models significantly underperform.

# G   TECHNICAL SETUP

All model prompting was done through the respective APIs of the models' parent company in Python. Models were run at least once in multiple-context with most models being run at least once in both single-context and multiple-context. After determining that there was no statistical difference between the two paradigms, any subsequent runs of models were run in multiple-context (due to the decreased cost of inference because of less input token usage).

Our exact runs of model paradigms are as follows:

- Claude Opus 4.1, multiple-context: 1
- Claude Sonnet 4, multiple-context: 1
- Claude Sonnet 4, single-context: 1
- Gemini 2.0 Flash, multiple-context: 1
- Gemini 2.0 Flash, single-context: 1
- Gemini 2.0 Flash images, multiple-context: 1
- Gemini 2.5 Pro, multiple-context: 1
- Gemini 2.5 Pro, single-context: 1
- ChatGPT-4o, multiple-context: 1
- ChatGPT-4o, single-context: 1
- GPT-4.1, multiple-context: 1
- GPT-4.1, single-context: 1
- GPT-4.1 with GPT-image-1, multiple-context: 1
- GPT-4.1 with GPT-image-1, single-context: 1
- GPT-5, high reasoning, multiple-context: 2
- GPT-5, medium reasoning, multiple-context: 1
- GPT-5, low reasoning, multiple-context: 1
- GPT-5, minimal reasoning, multiple-context: 1
- o3, high reasoning, multiple-context: 2
- o3, high reasoning, single-context: 1
- o3, medium reasoning, multiple-context: 1
- o3, low reasoning, multiple-context: 1
- o3-Pro, high reasoning, multiple-context: 2
- o3-Pro, high reasoning, single-context: 1
- o3 with GPT-image-1, high reasoning, multiple-context: 4
- o3 with GPT-image-1, medium reasoning, multiple-context: 1
- o4-mini, high reasoning, multiple-context: 1
- o4-mini, high reasoning, single-context: 1
- o4-mini, medium reasoning, multiple-context: 1
- o4-mini, medium reasoning, single-context: 1
- Gemini 3 Pro, high thinking, temperature 1.0, multiple-context: 1
- Gemini 3 Pro, high thinking, temperature 0.55, multiple-context: 1
- Gemini 3 Pro, high thinking, temperature 0.1, multiple-context: 1
- DeepSeek R1, default reasoning, multiple-context: 1
- Qwen 3, default reasoning, multiple-context: 1
- Qwen 3 VL, default reasoning, multiple-context: 1
- gpt-oss-120b, high reasoning, multiple-context: 1

The strongest OpenAI reasoning models were run a single additional time with variations on the 'reasoning_effort' parameter as described in Appendix E.1. Our reported data in all other sections is based on the 'reasoning_effort' parameter being set to 'high'.

For all models which allowed modification in their 'temperature' parameter, we set it to 0.1 (reasoning models from OpenAI and Anthropic do not allow modification to temperature). All other hyperparameters were left at their default value.

For OpenAI, we used the openai package, version 1.76.2; for Gemini, the google-genai package version 1.14.0 (for Gemini 3 Pro this was updated to 1.51.0); for Claude, the anthropic package version 0.52.2. We used the Miniconda environment manager to create a Python 3.12.9 environment, and the complete frozen package list ('python_env.yml') is available in our repository (see Appendix B).

For open models (including gpt-oss-120b) we used the openai package (same version as before) and the API platform OpenRouter.

Our prompt processing pipeline was run on a 2019 Razer Blade Stealth running Arch Linux with 16 GB of RAM, an 8 core Intel CPU (i7-8565U), and no dedicated GPU.

All data collected from models totaled around $1000 US in cost. The exact versions of models is listed in Table S2.

Data pre-processing was performed in a Jupyter Notebook file. We provide .md and .html versions of the output in addition to the raw file for convenience.

Statistical analyses were performed using R 4.3.3 in PyCharm with the R plugin installed. Miniconda was also used to create a replicable environment and the frozen package list ('r_env.yml') is available in our repository. Our analyses were performed within an R Markdown script. We provide .html and .pdf exports for simplified viewing.

# H   SUPPLEMENTAL FIGURES

| | Step 1 | Step 2 | Step 3 | Step 4 | Result |
|---|---|---|---|---|---|
| Instructions | Generate an image with two capital letter "B" next to each other. | From there, modify the image with the left figure mirrored so that it points left. | From there, modify the image with the two figures aligned such that the two vertical lines are overlaid. | From there, modify the image with a lowercase letter "v" affixed to the middle of the top of the figure. | Canonical form: Butterfly |
| Mental Image | **BB** | **ꓭB** | **ꓭB** | **ꓭB** | "Butterfly" |

Figure S1: An example image generation output from GPT-image-1 in combination with o3. No seed image was given to the model, but subsequent steps retained the previous image and asked for its modification.

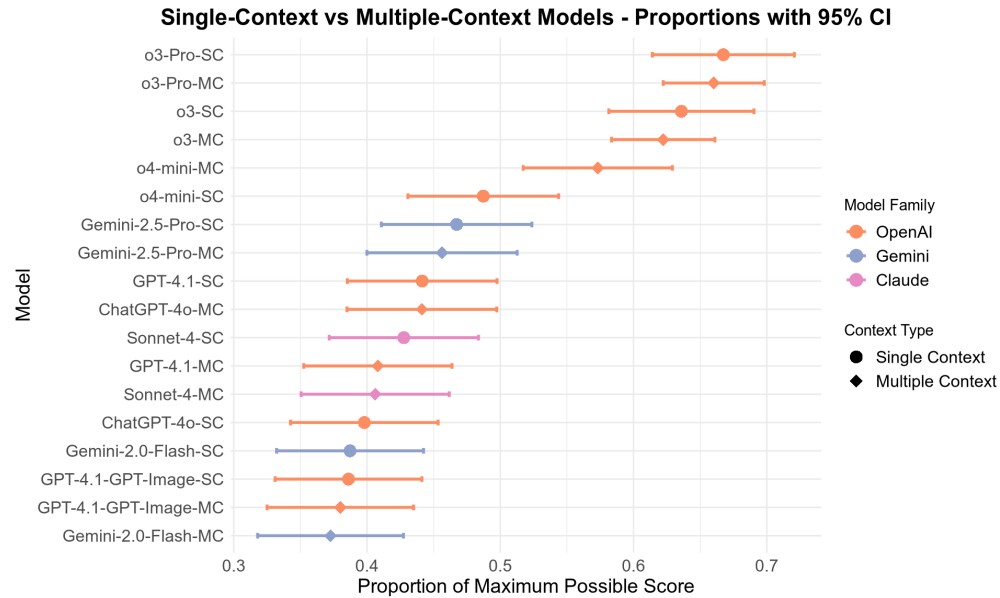

Figure S2: 95% confidence intervals showing the differences between Single-Context and Multiple-Context. Single-Context has a slight, non-significant edge in most cases. All context variant comparisons were non-significant after correcting for multiple-comparisons.

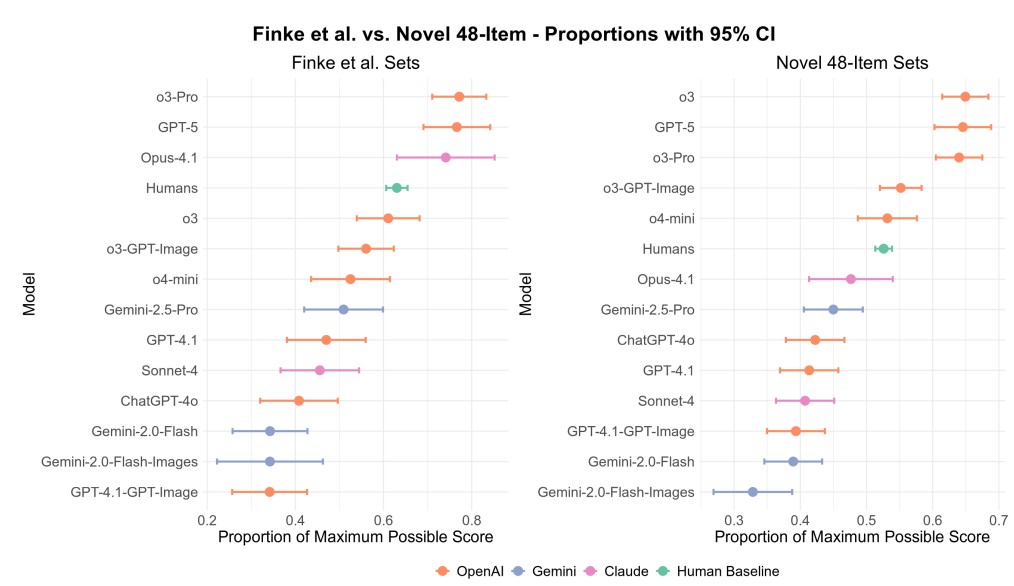

Figure S3: Separated 95% proportions of maximum possible score between Finke et al. Items (left) and 48 Novel Items (right).

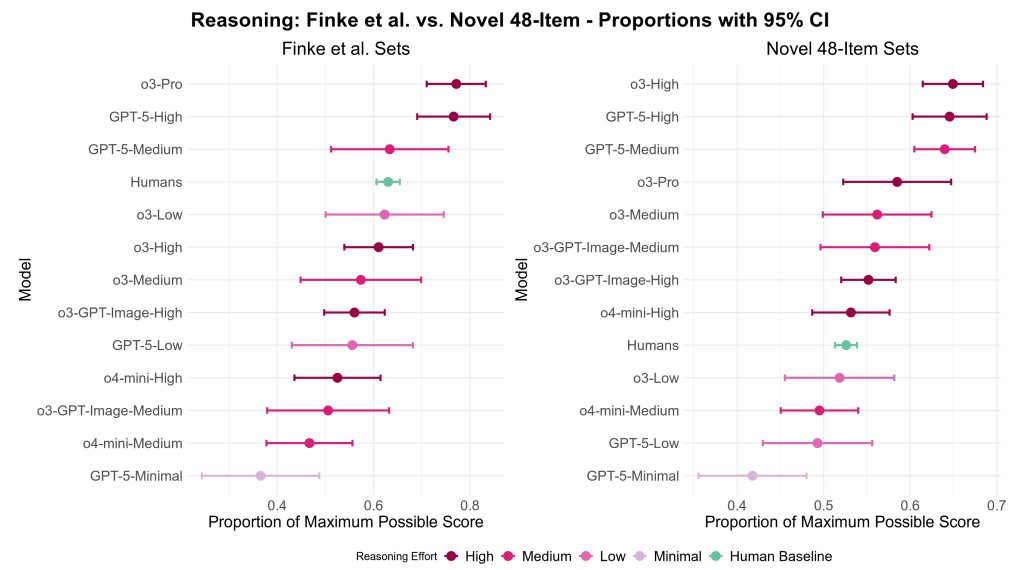

Figure S4: Hyperparameter comparison of results (95% confidence intervals of proportions of maximum possible score) after modifying 'reasoning_effort' parameter in OpenAI Large Language Reasoning Models. Human baseline is included. Generally, as reasoning effort (token generation amount) increases, so does performance. Results are separated by Finke et al. Items (left) and 48 Novel Items (right).

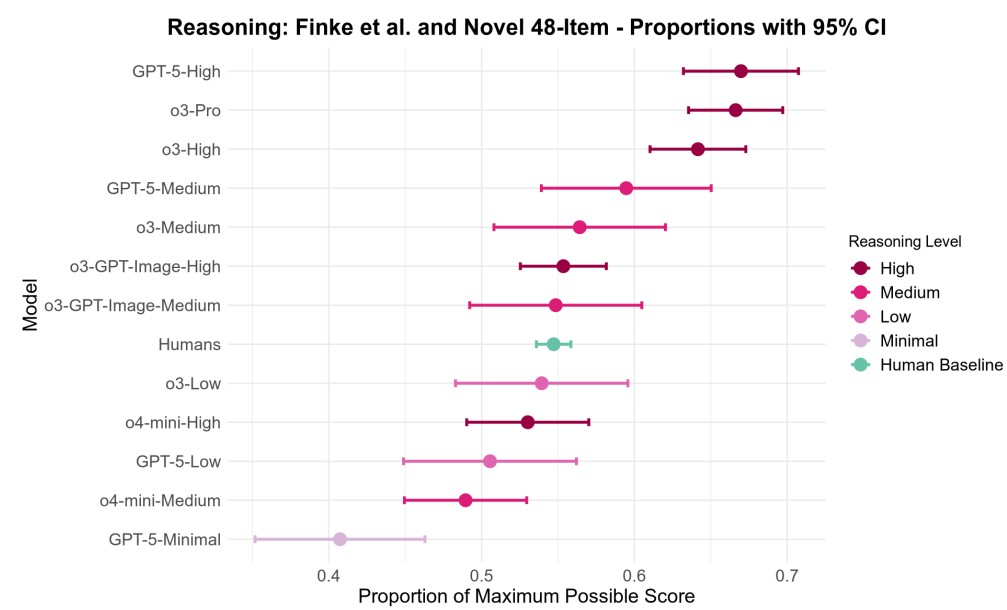

Figure S5: Hyperparameter comparison of collapsed results (95% confidence intervals of proportions of maximum possible score) after modifying 'reasoning_effort'.

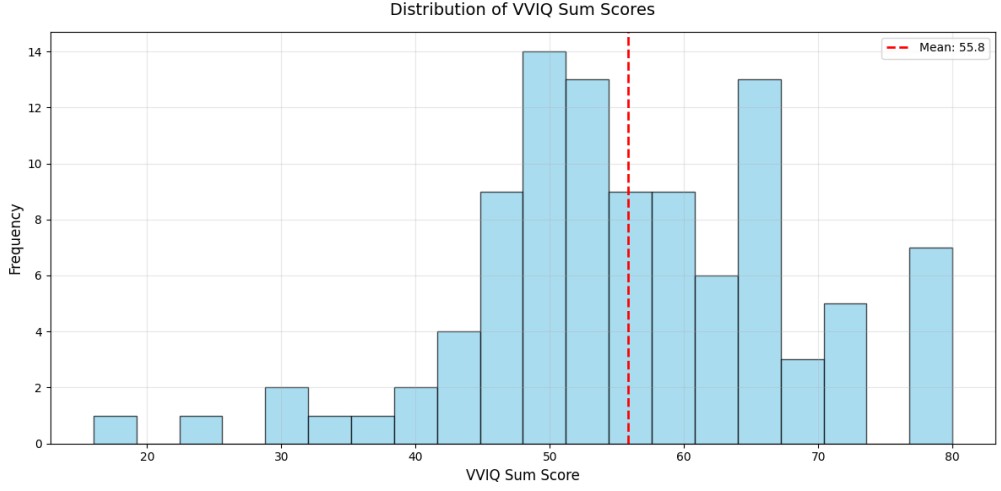

Figure S6: The distribution of VVIQ scores for our 100 human subjects. 1 subject qualified as an aphantasic under the strictest condition (VVIQ = 16). The characteristic left skew of the distribution is clear.

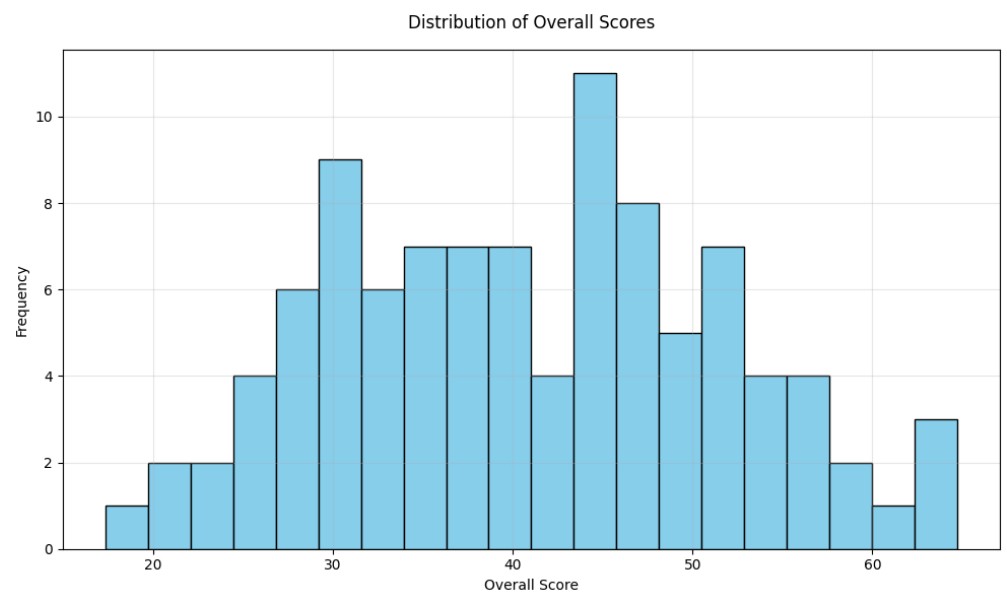

Figure S7: The distribution of overall graded scores for the 100 human subjects. As the subset of questions given to each subject was random, the overall difficulty of each set was not guaranteed and therefore some noise in the score distribution is expected.

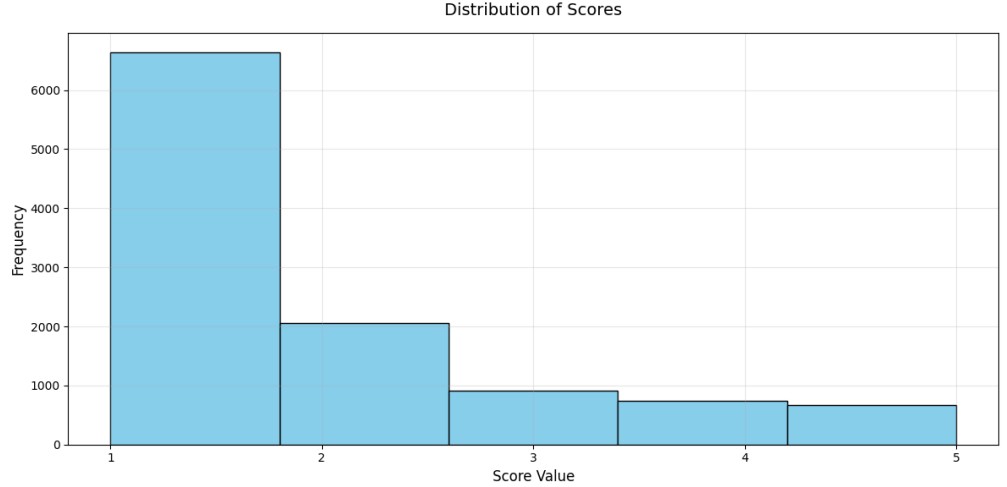

Figure S8: The distribution of scores given to unique answers by the 376 crowd-sourced subjects. ≈5 per each of the 1911 unique responses. Experts gave more high ratings than the random subjects.

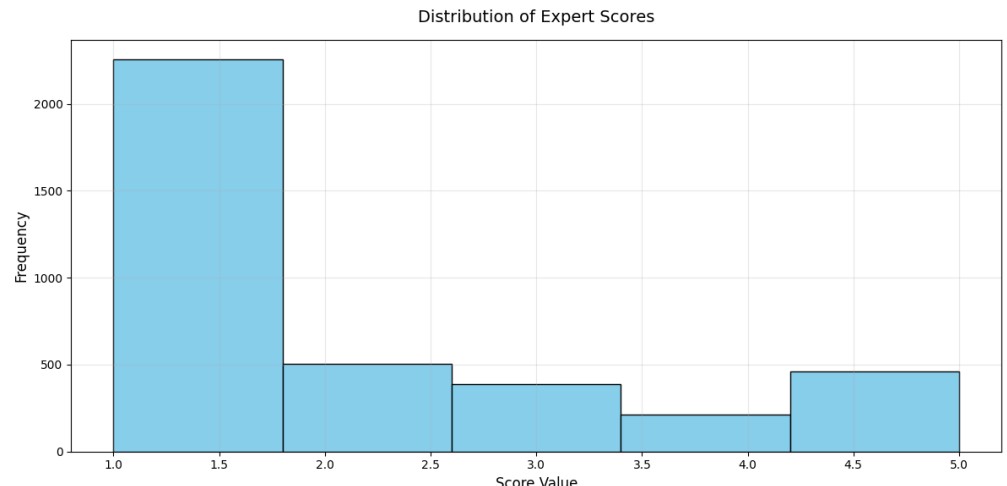

Figure S9: The distribution of scores given to unique answers by the 2 expert subjects. 2 to each of the 1911 unique responses.

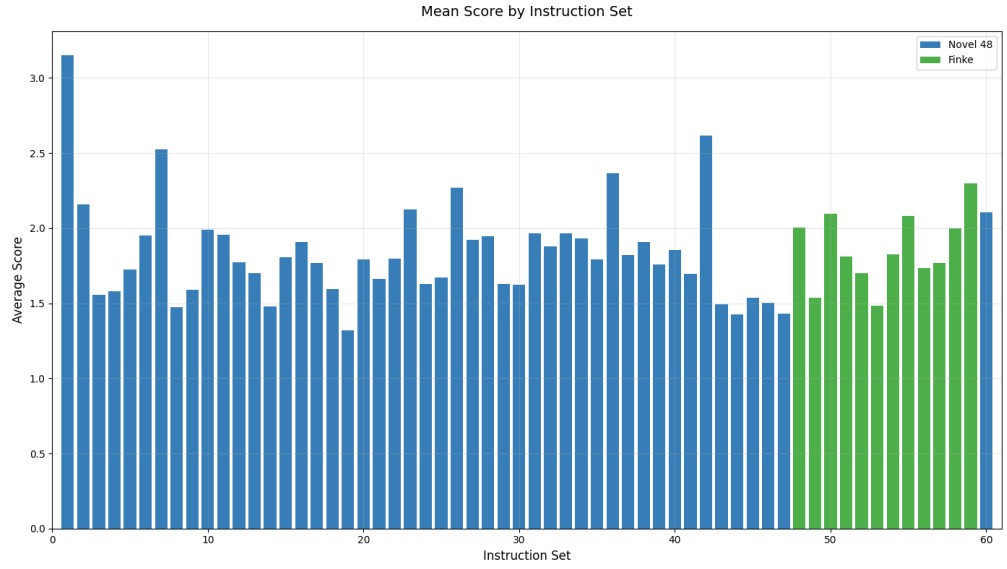

Figure S10: The mean crowd-sourced score for each answer given in each instruction set. Particularly difficult trials had a large variety of answers, many with lower scores.

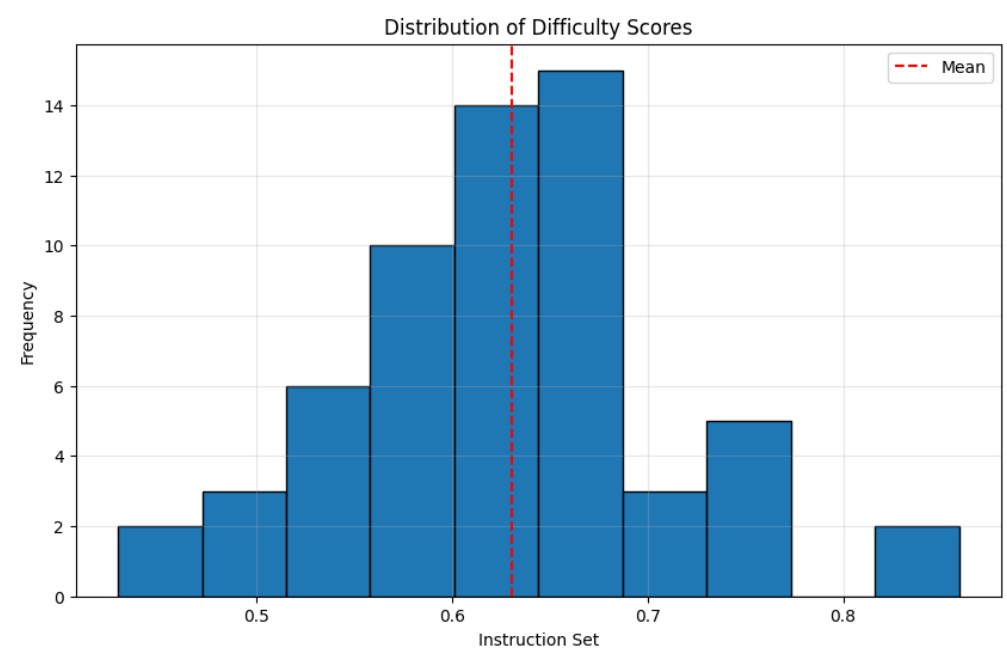

Figure S11: Distribution of calculated difficulty scores per instruction set. Our instruction sets showed strong variance of difficulty.

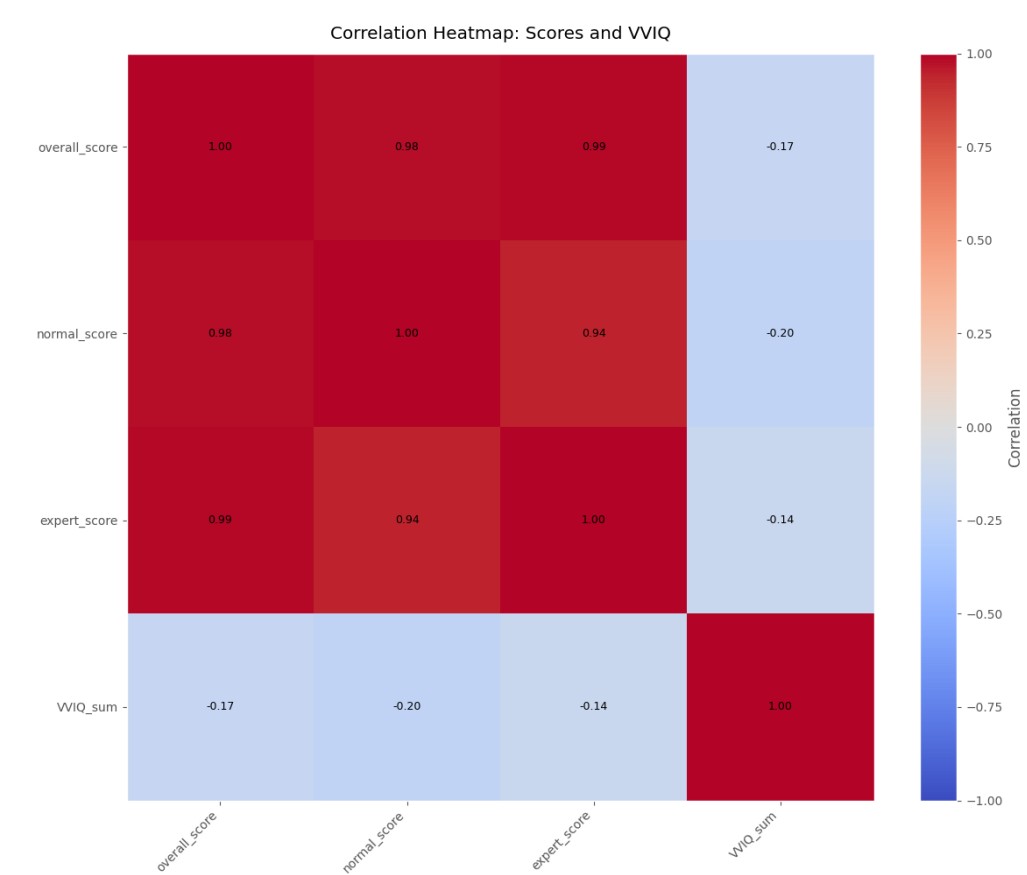

Figure S12: Correlation between VVIQ score sum, 'overall_score', 'normal_score', and 'expert_score' (see Appendix D.1 for terminology explanation).

# I  SUPPLEMENTAL TABLES

Table S1: Model selection and features. For OpenAI reasoning models, 'reasoning_effort' was set to 'high' (as shown) and, for the models indicated, GPT-image-1 was integrated. For Gemini models, default parameters regarding reasoning were retained and native image generation tools were used. For Claude models a reasoning token budget was manually given (well above what was ever allocated). This table only shows the primarily graded models, not the reasoning variations in our reasoning model effort analysis.

| Model | Reasoning | Image Generation |
|---|---|---|
| GPT-5 | High | No |
| o3-Pro | High | No |
| o3 | High | Yes |
| o4-mini | High | No |
| ChatGPT-4o | No | No |
| GPT-4.1 | No | Yes |
| Gemini 2.5 Pro | Dynamic | No |
| Gemini 2.0 Flash | No | Yes |
| Claude Sonnet 4 | 4000t | No |
| Claude Opus 4.1 | 9000t | No |

Table S2: Model Versions. Model versions used in our analysis. When possible exact dated versions are given, if no such version is available the date of usage was also provided.

| Model | Version |
|---|---|
| GPT-5 | gpt-5-2025-08-07 |
| o3 | o3-2025-04-16 |
| o3-Pro | o3-pro-2025-06-10 |
| o4-mini | o4-mini-2025-04-16 |
| ChatGPT-4o | chatgpt-4o-latest (July 2025) |
| GPT-4.1 | gpt-4_1-2025-04-14 |
| GPT-image-1 | gpt-image-1-2025-04-23 |
| Gemini 2.0 Flash | gemini-2.0-flash (February 2025) |
| 2.0 Flash Image Preview | gemini-2.0-flash-preview-image-generation (May 2025) |
| Gemini 2.5 Pro | gemini-2.5-pro-preview-05-06 |
| Claude Sonnet 4 | claude-sonnet-4-20250514 |
| Claude Opus 4.1 | claude-opus-4-1-20250805 |
| Gemini 3 Pro | gemini-3-pro-preview (November 2025) |
| DeepSeek R1 | deepseek-r1-0528 |
| Qwen 3 | qwen3-235b-a22b-thinking-2507 |
| Qwen 3 VL | qwen3-vl-235b-a22b-thinking |
| gpt-oss-120b | gpt-oss-120b |

Table S3: Reasoning Variations. Reasoning level variations used for reasoning analysis.

| Model | Reasoning Levels Used |
|---|---|
| GPT-5 | Minimal, Low, Medium, High |
| o3-Pro | High |
| o3 | Low, Medium, High |
| o3 with GPT-image-1 | Medium, High |
| o4-mini | Medium, High |

Table S4: Results Breakdown. Models in bold indicate highest performer in the group. The human baseline is in green. Models in purple surpass the human baseline significantly [** $p < .01$, *** $p < .001$]. Models in blue are not significantly [ns] different from the human baseline (i.e., at human level).

| Agent | Score | Max Possible Score | Proportion |
|---|---|---|---|
| *Finke et al. Items* | | | |
| Humans | 961.096 | 1525 | 0.6302 |
| o3[ns] | 109.900 | 180 | 0.6106 |
| o3 + GPT-image-1 | 134.483 | 240 | 0.5603 |
| **o3-Pro**[***] | **138.908** | **180** | **0.7717** |
| GPT-4.1 | 56.407 | 120 | 0.4701 |
| GPT-4.1 + GPT-image-1 | 41.000 | 120 | 0.3417 |
| o4-mini | 63.008 | 120 | 0.5251 |
| Gemini 2.5 Pro | 61.125 | 120 | 0.5094 |
| Gemini 2.0 Flash | 41.100 | 120 | 0.3425 |
| Gemini 2.0 Flash + Images | 20.538 | 60 | 0.3423 |
| Claude Sonnet 4 | 54.652 | 120 | 0.4554 |
| Claude Opus 4.1[ns] | 44.467 | 60 | 0.7411 |
| GPT-5[**] | 91.950 | 120 | 0.7663 |

Table S5: Novel 48-Item Expansion Results Breakdown. Models in bold indicate highest performer in the group. The human baseline is in green. Models in purple surpass the human baseline significantly [** $p < .01$, *** $p < .001$]. Models in blue are not significantly [ns] different from the human baseline (i.e., at human level).

| Agent | Score | Max Possible Score | Proportion |
|---|---|---|---|
| *48 Novel Items* | | | |
| Humans | 3137.120 | 5965 | 0.5259 |
| **o3**[***] | **467.490** | **720** | **0.6493** |
| o3 + GPT-image-1[ns] | 529.699 | 960 | 0.5518 |
| o3-Pro[***] | 460.713 | 720 | 0.6399 |
| GPT-4.1 | 198.455 | 480 | 0.4134 |
| GPT-4.1 + GPT-image-1 | 188.827 | 480 | 0.3934 |
| o4-mini[ns] | 255.123 | 480 | 0.5315 |
| Gemini 2.5 Pro | 215.949 | 480 | 0.4499 |
| Gemini 2.0 Flash | 186.882 | 480 | 0.3893 |
| Gemini 2.0 Flash + Images | 78.807 | 240 | 0.3284 |
| Claude Sonnet 4 | 195.488 | 480 | 0.4073 |
| Claude Opus 4.1[ns] | 114.352 | 240 | 0.4765 |
| GPT-5[***] | 309.873 | 480 | 0.6456 |

