# OpenReview forum: "Artificial Phantasia: Evidence for Propositional Reasoning-Based Mental Imagery in Large Language Models"
_ICLR.cc/2026/Conference — Submitted to ICLR 2026_

### Official Review · Reviewer_UkQT · 2025-10-30

**Soundness:** 2
**Presentation:** 3
**Contribution:** 2
**Rating:** 2
**Confidence:** 5

**Summary:**

This paper introduces a novel benchmark for assessing complex cognitive behavior in Large Language Models through a classic mental imagery tasks. By comparing several state-of-the-art LLMs with 100 human participants, the authors find that the best models surpass average human performance despite lacking any visual architecture. The results suggest that LLMs may possess emergent reasoning abilities sufficient to solve imagery-dependent tasks using purely linguistic representations.

**Strengths:**

- Adapting a classical mental imagery paradigm to test LLMs represents a  step toward evaluating higher-order cognition in LLMs.
- The finding that leading LLMs outperform the human baseline on the mental rotation task previously thought to require visual imagery is interesting.

**Weaknesses:**

The paper’s central assumption—that the inability to generate images directly implies a lack of image-rendering capability—appears conceptually overstated. Prior work [1] has shown that large language models can acquire substantial visual priors from purely linguistic training, sometimes even displaying visual reasoning or “hacked” image understanding. The fact that a model cannot explicitly output an image does not necessarily mean it fails to internally “render” or simulate a scene during reasoning [2]. Humans, for instance, often visualize mental imagery without ever producing drawings.

The claim would be more convincing if supported by evidence from representation-level analysis rather than behavioral absence. Specifically, extracting hidden states and attempting to reconstruct images could provide direct support for the hypothesis.

Furthermore, all experiments are performed on closed-source models, which limits reproducibility and interpretability. The appendix briefly mentions open-source models, but no concrete results or analyses are provided.

[1] Han, J., Tong, S., Fan, D., Ren, Y., Sinha, K., Torr, P. and Kokkinos, F., 2025. Learning to See Before Seeing: Demystifying LLM Visual Priors from Language Pre-training. arXiv preprint arXiv:2509.26625.

[2] Luo, Dezhi, Qingying Gao, and Hokin Deng. "Rethinking the Simulation vs. Rendering Dichotomy: No Free Lunch in Spatial World Modelling." arXiv preprint arXiv:2510.20835 (2025).

**Questions:**

- Consider conducting representation-probing experiments—for instance, extracting hidden states and testing whether they contain decodable visual information or spatial priors, whether they can lead to interpretable and reasonable image.
- Include open-source baselines to improve reproducibility and interpretability, ideally with side-by-side analyses.
- Clarify the conceptual distinction between image generation and image rendering, and explicitly discuss whether implicit rendering might occur during multimodal reasoning even in text-only models.

---

> ### Author Response · Authors · 2025-11-27
> **Reponse and Concerns Re: Review**
>
> Overall this review conceptually misunderstands the paper. First, the task itself is not a mental rotation task; mental rotation is a key part of some of the instruction sets, but this is an oversimplification of the “image” reconstruction task. Furthermore, the “prior” work referenced, is not prior work. Han et al. was preprinted on September 30th, 2025, 6 days after the paper submission deadline. Lou et al. was preprinted on October 15th, 2025, 21 days after the paper submission deadline. Neither has appeared in an archival peer-reviewed venue.
>
> In any case, the claims in both preprints are irrelevant to our manuscript since they do not support the argument being made by the reviewer:
>
> - Han et al. makes claims about perceptual abilities of models. Of their three primary claims, two of them do not apply in this scenario at all: “data-efficient visual adaption” and “LLMs as strong vision encoders” both necessitate known, clear, visual tasks (i.e., pixels are presented to the model). The third claim, “programmatic visual knowledge”, reinforces our point that models can represent image-like data in a discursive format (code).
>
> - The argument the reviewer makes with regard to Luo et al. is circular. If we assume (not prove, as the paper does not provide empirical data) that mental imagery in humans is pictorial, then of course they will “render” images whether people are aware of them or not. The core assertion of our paper is that we cannot simply assume this and use LLMs as a useful tool to disentangle a question about formats of mental imagery that has proved to be really hard to address in humans. Furthermore, please see section 1.2 where we have already discussed this.

---

### Official Review · Reviewer_5Vv1 · 2025-10-30

**Soundness:** 2
**Presentation:** 2
**Contribution:** 2
**Rating:** 2
**Confidence:** 3

**Summary:**

The paper investigates whether LLms can solve tasks traditionally requiring visual imagination. Using newly designed “mental imagery” problems and human benchmarks, the authors show that LLMs (e.g., GPT-5, o3-Pro) outperform humans even without visual inputs, suggesting that abstract propositional reasoning can account for their success.

**Strengths:**

- interesting study of pictorial vs propositional representation in LLms.

- Good dataset with clear methodology.

- The experiments show modern reasoning-tuned LLMs exceed in reasoning.

**Weaknesses:**

- the scoring validity is based on ordinal rate (1-5) may include a noise, how to treat that?

- the use of chi-square tests on “proportion-of-maximum scores” may violate the  independence and equal-variance assumptions

- the humain/LLM  protocol may produced biases results. Humans receive audio-only instructions while LLMs read text and can then rely on memory. Could you clarify this point?

- the chose of evaluation design : single-run; temperature = 0.1 and somtime = 1, etc are used without justification. Please clarify.

**Questions:**

see weaknesses

---

> ### Author Response · Authors · 2025-11-27
> **With regard to weakness one**
>
> We honestly are not sure what to make of this comment since the reviewer did not elaborate on their concern. Ordinal rating scales are par for the course in psychological research and we abided to this extremely standard rating methodology.  All data collected from models and humans is of the same form, therefore any potential measuring noise (if that is what the reviewer is concerned with) should be similar across both paradigms.

---

> ### Author Response · Authors · 2025-11-27
> **With regard to weakness two**
>
> All chi-squares values were obtained by comparing data from two different groups (e.g., humans vs a specific model). So we believe there is no risk of violating the assumption of independence. Pearson's Chi-square tests do not make assumptions about equal variance.

---

> ### Author Response · Authors · 2025-11-27
> **With regard to weakness three**
>
> Presenting text to humans necessarily introduces visual priors which are not introduced to language models as the text is not provided as an image. Furthermore, providing audio to models introduces noise as, to the best of our understanding, models primarily use transcription to present the language back to the models. LLMs are not reading text in the same way humans would.

---

> ### Author Response · Authors · 2025-11-27
> **With regard to weakness four**
>
> Models were typically run a single time with a temperature of 0.1 to reduce randomness of their outputs (and thus providing more valid responses to the one-shot paradigm). This is the case for all non-reasoning models and Gemini 2.5 Pro. Reasoning models are required by their production companies to have higher temperature parameters and, therefore, were left at default (as they cannot be changed). To test the impact of this, we have just run Gemini 3 Pro with temperature values 0.1, 0.55, and 1.0. These results are still being evaluated and will be provided shortly.  A preliminary (pre-crowdsourcing) view of these results implies, to us, that temperature has had no significant effect. We did not run models a large number of iterations due to the cost of data collection and the diminishing returns of repeated iteration. While more iterations would have improved the significance of results, we have shown our strongest results to be well over the necessary threshold.

---

> > ### Author Response · Authors · 2025-12-03
> > **Gemini 3 Pro Temperature Analysis Results**
> >
> > When testing Gemini 3 Pro with a single iteration with temperature 0.1, 0.55, and 1.0 (to test the range of values nearest to what we used and in the middle), we saw no significant relationship between the hyperparameter and results. Please see our revisions to the paper where we discuss this.

---

> ### Author Response · Authors · 2025-11-27
> **General Response to Reviewer**
>
> Given the simplicity of the points provided and lack of further concerns, we kindly urge the reviewer to update their score to something more fitting to the appeal of our paper.

---

### Official Review · Reviewer_1U8T · 2025-11-01

**Soundness:** 1
**Presentation:** 2
**Contribution:** 2
**Rating:** 2
**Confidence:** 3

**Summary:**

This paper introduces a classic cognitive task to benchmark complex mental imagery in large language models (LLMs). The authors created 48 test points (with 12 existing test points) based on a classic object reconstruction paradigm from cognitive psychology, where participants/models follow multi-step verbal/textual instructions to transform letters and shapes and then identify the resulting object. The results find that the best LLMs significantly outperformed the human baseline, with performance improving as more reasoning tokens were allocated. This suggests that LLMs can solve tasks traditionally thought to require pictorial mental imagery through purely propositional, language-based reasoning.

**Strengths:**

1. This paper investigates an interesting research question: can LLMs/LRMs solve propositional reasoning-based mental imagery problems. It provides empirical results on investigating how LLMs perceive and manipulate visual-related operations.
2. This paper offers detailed background, analysis and discussion from the cognitive science and mental imagery perspective, which makes this paper more complete.

**Weaknesses:**

1. The claim and conclusion are not well supported. Since this problem type is not new, it cannot be concluded that the evaluated tasks are 'ex novo'. SOTA LLMs may be trained on problems within the same distribution so that they achieve a superior performance. Additionally, the evaluation is only based on this problem type, instead of a series of tasks involving propositional reasoning and mental imagery (such as Connect the dots), which weakens the soundness of this paper.
2. The conclusion from the discussion is not supported by empirical evidence. There is no analysis of model internals (e.g., attention patterns, activation vectors, or reasoning traces) that could substantiate how the models arrive at correct answers.

**Questions:**

1. Could the authors explain more about why humans are provided with the audio instead of the text input (which maintains the same setting as the models)?
2. See weakness 1 above.

---

> ### Author Response · Authors · 2025-11-27
> **With regard to weakness one and question two**
>
> There is some probability of the original 12 items in the Finke et al. task being in the training data. The original paper that devised the *type* of task that we used in our study is over 35 years old (Finke et al., 1989) and it has been cited more than 400 times. We, of course, knew this when designing our study and we are not particularly worried about it for the following reasons. First, only 12 out of 60 instruction sets appeared in the 1989 paper; the rest of them are completely novel designs. Second, the novel items include rather idiosyncratic examples that would be astounding if anything like them were in the training data (e.g., a computer mouse made with an S, a C, a horizontal line, and a D). Third, and related to the previous point, even if the *type* of task is present in the training data, and even if LLMs have information about the shape of letters (e.g., they know an o is like a circle and they know ice cream can be drawn with circles), the models still have a challenging task of compositionality: put together all the elements, in the right ***spatial*** layout, in some cases after 4 transformations including rotation and mirroring, and then use the resulting shape to read out the actual answer. This demanding task goes well beyond knowing that ice cream icons have circles (notoriously, an enormous amount of other things can also be drawn using circles). Fourth, no model, not even the best performing ones, universally solved all the original Finke items, suggesting that the models are not simply finding a ready-made solution from their training data. Finally, in the case of some models (e.g., GPT 4.1 and o3) we have reasons to believe that they may share their training data (both have a June 01, 2024 knowledge cutoff). Even if some information relevant to our task were buried in the models' training data, it is the parameters and general architecture of the models that ultimately determine how well they were able to find relevant information and solve our novel items. Thus, our data shows that some advanced LLMs can reason about composite shapes and their spatial layout by relying solely on linguistic embeddings in ways that seriously challenge the hypothesis that a pictorial format is necessary for completing our task.
>
> Furthermore, with regard to "connect the dots": this task has no basis in the cognitive science literature as a mental imagery task and would not be adequate to test our hypothesis about mental imagery and propositional formats.

---

> ### Author Response · Authors · 2025-11-27
> **With regard to weakness two**
>
> Our work provides a foundation study of an emergent cognitive ability. We are currently analyzing reasoning traces from OpenAI’s gpt-oss 120b, DeepSeek R1, and Qwen 3, but lower level representations analyses are much more akin to neuroscience while we are discovering and analyzing behavior. While future studies should perform more mechanistic analyses, no literature (aside from ours) has documented these emergent capabilities. The scope of this project does not necessitate these low-level analyses as they, as a result, would stand entirely on their own, and no study would be able to perform them without what our work has documented.

---

> ### Author Response · Authors · 2025-11-27
> **With regard to question one**
>
> Presenting text to humans necessarily introduces visual priors which are not introduced to language models as the text is not provided as an image. Furthermore, providing audio to models introduces noise as, to the best of our understanding, models primarily use transcription to present the language back to the models.

---

### Official Review · Reviewer_Gt5a · 2025-11-01

**Soundness:** 4
**Presentation:** 4
**Contribution:** 4
**Rating:** 8
**Confidence:** 5

**Summary:**

This work investigates the emergent ability of LLMs to perform tasks that are thought to depend on mental imagery. The results show surprisingly strong performance, for some models even surpassing a human baseline, despite the fact that language models are only trained on text and therefore putatively unable to engage in imagistic reasoning. These results make an interesting contribution to classic debates in cognitive science, and also contribute a novel benchmark on an as yet unexplored task domain for LLMs.

**Strengths:**

- The results provide a clear example of how studying LLMs can make an interesting and surprising contribution to debates in cognitive science.
- The proposed benchmark tests LLM capabilities in a new domain, and there is still room for improvement so the benchmark will likely be useful for future studies as well.
- A range of different models are tested, including both base models and reasoning-augmented models.
- The writing is clear and there is a nice review of the relevant literature in cognitive science.
- The dataset involves newly constructed problems to avoid concerns about training data contamination.
- A publicly available code repo is included.
- All of the results are accompanied by appropriate error bars and statistical tests.
- Results are evaluated by human judges, allowing for assessment of open-ended responses.

**Weaknesses:**

- Can the distinction between imagistic and propositional solutions be formalized? And how do we know that the LLMs have not induced something like an imagistic internal model? Formalizing what it means for an internal representation to be imagistic rather than propositional would help to answer this question. Although text-only LLMs are not trained on images, in principle this doesn't preclude them from inducing an internal model that has imagistic properties, as this may be a useful latent variable for predicting text-based descriptions of visual properties. One way to formalize this might be in terms of representational similarity analysis, i.e. an internal representation might be considered imagistic if it has the same similarity structure as actual images. In the discussion, the authors described a potential pseudo-imagistic solution involving spatial relations, but I still found myself wondering what makes a representation / solution imagistic per se. Some additional discussion of this issue might be helpful for guiding future research that could more directly investigate the internal solutions that these models use.
- In some parts of the paper it's claimed that the proposed dataset is uniquely immune to concerns about data contamination because it tests a putatively non-verbal capacity. I found this unconvincing for two reasons. First, there are many studies that investigate emergent abilities in LLMs by proposing completely new datasets and tasks. In these cases, it is not plausible that the same problems simply happen to have already been present on the internet. Second, although the task studied in this work has been previously proposed to rely on images, the task is still defined in terms of text-based inputs and outputs. Therefore, there is no reason in principle that it can't have been present in the training data even if the training data is purely text-based (I do not think this is a concern because of the first reason, but I think there are plenty of other studies that are equally immune to this concern because of the use of novel tasks).
- The dataset should be useful for future studies, but given that the dataset is now publicly available this raises concerns about training data contamination for future models. Is it possible to automate the generation of new problems so that future models can easily be tested on problems that cannot have been present in the training data?
- Only proprietary models are studied. It would be informative to include open-source models, particularly open-source reasoning models such as DeepSeek r1 that would allow direct inspection of the reasoning traces to better understand the model's solutions.
- How do the length of the reasoning traces relate to model accuracy, and to human performance? The effect of reasoning effort seems to suggest that longer reasoning traces help the models to perform better. It would be interesting to relate this to human performance, e.g. to determine whether the length of the reasoning traces is meaningfully correlated with human accuracy or human response times.
- Are any of the models trained on images, and is it possible to determine whether this impacts performance? One way to determine this might be to compare performance for an open-source VLM (e.g., Qwen) with the performance of the LLM base model prior to training on images.

**Questions:**

- Can the distinction between imagistic and propositional solutions be formalized, potentially allowing for future work that investigates the internal representations used to solve this task?
- Does the content of the reasoning traces reveal anything interesting about the solution used to solve the task?
- Does the length of the reasoning traces correlate with human accuracy or response times?
- How does training on images impact performance on the task?
- Is there any research that tests congenitally blind individuals on this task?

---

> ### Author Response · Authors · 2025-11-27
>
> Dear reviewer,
>
> We deeply appreciate your thoughtful review and are still working on our responses. We will post them within the next day or two, and we additionally hope to have further results to share shortly after.
>
> Best,
>
> The authors

---

> ### Author Response · Authors · 2025-12-03
> **With regard to the formalization of the difference between imagistic and propositional solutions**
>
> Thanks for pressing us to be more explicit about these two different formats of representation. By format of representation we do not mean “what” information is being stored and manipulated by a cognitive system (be a human or a machine), in other words, we are not concerned with the content of the representation. Rather, we care about “how” information is represented, its format. Propositional formats use only symbolic elements (and syntax) to represent information. These elements are discrete and can typically be combined to create new representations. Language is the prototypical example of propositional representation. The words “house” and “red” are symbolic, each have whole meanings and if combined they can create complex representations in virtue of their individual meanings like when we say “red house”. In contrast, imagistic (or depictive or pictorial) representations are not like this. They do not have a “canonical decomposition”, in other words, their constituent parts are not necessarily meaningful/representational and the complex information that they encode is not the result of combining atomic parts. Individual strokes in Mona Lisa’s face are not independently meaningful from each other, and there are no obvious ways of parsing the Mona Lisa’s strokes (unlike with the words “red” and “house” in “red house”). Imagistic representations are spatially continuous and have an analogue format.
>
> Translating this old distinction from cognitive science into machine learning, we think that the properties of a propositional vs a depictive format of representation should be detectable in both the LLMs’s behavior and in the model’s weights and embeddings. For the latter, a mechanistic approach would be required and, as research with DNNs has shown, patterns of activity in early layers can be quite similar to patterns of activity in early visual cortex. This information is spatially encoded (retinotopically in the case of humans) and it does not allow for a canonical decomposition (i.e., information is continuous rather than discrete). For the former, we think that our task is one way (of many) to infer what the format of representation must be. Given how the LLMs that we tested are trained and how they work (see other replies), we can expect that their format of representation is propositional (i.e., there are discrete, symbolic elements—tokens—that can be combined systematically to form representations). Our task is meant to show, then, that non-depictive, propositional representations are sufficient for performing certain tasks thought to require depictive representations.

---

> ### Author Response · Authors · 2025-12-03
> **With regard to the content of the reasoning traces**
>
> Unfortunately, due to only having access to the reasoning traces of open models. Our analysis has been inconclusive due to the extremely poor performance of such models. Please see our other responses regarding open models and our updated results.
>
> We have provided the reasoning traces (from the open models) in our updated repository.

---

> ### Author Response · Authors · 2025-12-03
> **With regard to the length of reasoning traces versus response time**
>
> We have taken a general look at the reasoning traces we have seen so far (and some assumptions we can make given the output length of the proprietary ones), and while the models can perform this task at a level above human performance, we believe comparing the length of the traces to human response times is a bit of a false equivalency. The best models consistently used over 5000 and up to 20000 tokens within the reasoning responses which, despite the slowness of responses, is orders of magnitude faster (and more content) than a human (Zheng and Meister, 2025). Humans took a median time of ~30 minutes to complete 15 instruction sets (and as little as 15 minutes, but no more than 1 hour and 15 minutes) while even the fastest models (o4-mini) took well over 3-5x that time while outputting far more data than a human ever could. Given the amount of data which LLMs can work with, it’s hard to say if we can draw any conclusions between human response time and LLM reasoning trace length. Finally, due to the online data collection paradigm we do not have fine-grained access to individual instruction response time. We only know the total time taken by humans (which could include breaks between instruction sets).
>
> Zheng, J., & Meister, M. (2025). The unbearable slowness of being: Why do we live at 10 bits/s?. Neuron, 113(2), 192-204.

---

> ### Author Response · Authors · 2025-12-03
> **With regard to the length reasoning traces versus response accuracy**
>
> While looking at the lengths of the reasoning traces given by the proprietary models (i.e., the token counts), we do see some trends between the length of the traces and the difficulty of the task (e.g., the computer mouse tasks, which both humans and the models routinely failed at). We consistently see long reasoning traces for the more difficult tasks regardless of the performance of the models.

---

> ### Author Response · Authors · 2025-12-03
> **With regard to image training**
>
> While we do not know the training data of the proprietary models, we can make some assumptions about how this may affect our results. Presuming that any vision models are trained by aligning a vision encoder towards the language decoder (and then unfreezing the weights of the full model and continuing training), there may be one key advantage of this: learning a linguistic binding for images. As this paradigm necessarily aligns images to a textual domain (i.e., the input to the text decoder), the model must learn how to represent images textually. These textual representations of images should be decomposable to a degree in which an image would not. This would provide a strong benefit to the propositional-reasoning case. Given human priors on visual or spatial information, our propositions will necessarily be influenced by perception, even in cases without perception.
> Unfortunately, the lack of knowledge regarding the training of proprietary models, as well as the extremely poor performance of the open weight models makes it very difficult to know exactly what is happening under the hood.
>
> For our analysis of open weight models, we did look at Qwen 3 and Qwen 3 VL with thinking. Both had very poor performance (Qwen 3 was the lowest scoring model of all models we tested), but Qwen 3 VL did outperform Qwen 3 (though, after accounting for multiple comparisons, non-significantly). This may give some evidence that learning textually grounded visual priors can improve the results (note that the image encoder should not be used in this case), however, we cannot assume these results are extensible to the proprietary models that we tested as the performance of Qwen was so severely worse.

---

> ### Author Response · Authors · 2025-12-03
> **With regard to the congenitally blind**
>
> This is a very interesting question, and while we do not believe that this task in particular has been studied in the congenitally blind, there are reports that the blind do have mental imagery and are able to complete tasks grounded within it (e.g., mental rotation of imagined objects, comparing the shapes of imagined objects, etc.; see Pylyshyn, 2002 for discussion). However, the Finke et al. (1989) task has been used in a patient with perceptual disorders (visual agnosia—inability to visually recognize objects after a motor accident and apparent thinning of visual cortex). This patient was able to correctly identify in mental imagery 11/12 of the Finke items and the 12th too after drawing it (incredibly, he was unable to visually recognize the object in his own drawing when shown to him later) (Behrmann, Moscovitch and Winocur, 1994). More generally and to the reviewer’s point, the fact that congenitally blind people can form mental images raises a question about the grounding of imagery in perception (maybe non-visual perception such as touch and proprioception provides enough grounding for completing this task). Furthermore, we have not found any prior work on finding individuals who are congenitally both aphantasic and blind, but these may be the most interesting individuals to study.
>
> Pylyshyn, Z. W. (2002). Mental imagery: In search of a theory. Behavioral and brain sciences, 25(2), 157-182.
>
> Behrmann, M., Moscovitch, M., & Winocur, G. (1994). Intact visual imagery and impaired visual perception in a patient with visual agnosia. Journal of Experimental Psychology: Human Perception and Performance, 20(5), 1068.

---

> ### Author Response · Authors · 2025-12-03
> **With regard to the questions of data contamination on the current study**
>
> There is some probability of the original 12 items in the Finke et al. task being in the training data. The original paper that devised the *type* of task that we used in our study is over 35 years old (Finke et al., 1989) and it has been cited more than 400 times. We, of course, knew this when designing our study and we are not particularly worried about it for the following reasons. First, only 12 out of 60 instruction sets appeared in the 1989 paper; the rest of them are completely novel designs. Second, the novel items include rather idiosyncratic examples that would be astounding if anything like them were in the training data (e.g., a computer mouse made with an S, a C, a horizontal line, and a D). Third, and related to the previous point, even if the *type* of task is present in the training data, and even if LLMs have information about the shape of letters (e.g., they know an o is like a circle and they know ice cream can be drawn with circles), the models still have a challenging task of compositionality: put together all the elements, in the right ***spatial*** layout, in some cases after 4 transformations including rotation and mirroring, and then use the resulting shape to read out the actual answer. This demanding task goes well beyond knowing that ice cream icons have circles (notoriously, an enormous amount of other things can also be drawn using circles). Fourth, no model, not even the best performing ones, universally solved all the original Finke items, suggesting that the models are not simply finding a ready-made solution from their training data. Finally, in the case of some models (e.g., GPT 4.1 and o3) we have reasons to believe that they may share their training data (both have a June 01, 2024 knowledge cutoff: https://platform.openai.com/docs/models/gpt-4.1 and https://platform.openai.com/docs/models/o3). Even if some information relevant to our task were buried in the models' training data, it is the parameters and general architecture of the models that ultimately determine how well they were able to find relevant information and solve our novel items. Thus, our data shows that some advanced LLMs can reason about composite shapes and their spatial layout by relying solely on linguistic embeddings in ways that seriously challenge the hypothesis that a pictorial format is necessary for completing our task.

---

> ### Author Response · Authors · 2025-12-03
> **With regard to the questions of data contamination for future studies**
>
> While we believe this is a critical concern, we wish for a different takeaway from the average reader than simply reproducing our results on every future model. Our findings have discovered that models can solve tasks which were thought to be entirely restricted towards the inherent multimodality of the human mind, in particular through multimodal imagery. Mental imagery, as a whole, is vastly understudied within AI systems. While there has been recent work on implementing something similar through architectural changes, no one has taken the step back to ask: “Do models already have something adjacent to mental imagery?”. As such, we are raising the case that, yes, models are capable of solving imagery-dependent tasks, and might have something that warrants this consideration. We hope that future researchers are able to design more tasks that accurately benchmark this phenomenon and provide a stronger picture of how these models are doing this. Critically, we do not wish to provide an actual benchmark at this time because of these issues. Furthermore, while we do provide our data, this is more for the replication of our results on the models we tested than solely replicating the paradigm endlessly with future model releases. We hope that strong enough open-source models (or the existing models later) are released to learn more about what is occurring in the near future to reduce the risk of training contamination (presuming that the reported training data cutoffs are accurate). But if this is not possible, we believe that what we have learned will inspire future researchers to evaluate models with different tasks which will be further removed from data contamination issues. Lastly, we are not certain of the impact of training on the task within the training data (as we discuss in the previous response).

---

> ### Author Response · Authors · 2025-12-03
> **With regard to open models:**
>
> Please see our general comment to all reviewers regarding our testing and analysis of open weight models (and our disappointment in their performance).

---

### Author Response · Authors · 2025-12-03
**General Comment to All Reviewers Regarding Open Models and Mechanistic Interpretation:**

We have completed analysis on 4 different open weight reasoning models: DeepSeek R1, Qwen 3, Qwen 3 VL, and gpt-oss-120b. The results were extremely disappointing. None of these four models came anywhere near human-level performance (nevermind to the level of the best performing proprietary models). As such there is no possibility of any mechanistic interpretation in our current task. We do not have access to *any* of the weights or reasoning traces from these proprietary models. We were surprised how poorly these models performed, and, unfortunately, we cannot make any strong conclusions as to *why* they perform so much weaker than their proprietary counterparts.

Notably, we tested Qwen 3 and Qwen 3 VL to see if there were any strong differences in performance between a strictly language model and a vision language model, but, after accounting for multiple comparisons, we found no significant difference. Qwen 3 VL *did* show a moderate improvement in performance (non-signficant), but both Qwen models were among the worst of all models we tested.

We provide the reasoning traces (and results) from the 4 models we tested in our updated repository if anyone wishes to analyze them in the future, but we believe any analysis of these cannot be extended to the strongest models.

---

### Author Response · Authors · 2025-12-03

We would like to thank everyone for their feedback on our paper, especially Reviewer Gt5a for their high-quality, detailed review.

We hope that the merits of our work can be seen clearly, especially considering their multidisciplinary impact.

We have provided a final, updated version of our paper in our most recent revision.

Thank you!

Best,

The authors

---

### Meta-Review · Area_Chair_2Zsz · 2026-01-05

**Summary:**

The papers presents a small-scale study on character-based mental imagery in (text-only) large language models. The results indicate that frontier models can recognize mental images (described in text). The authors present this as a surprising finding and show how the best models are able to do these tasks as well as humans.

The reviews are mixed. Before the rebuttal three reviewers clearly want to reject the paper, one votes for acceptance. The reasons for acceptance are

Reviewer Gt5a
> The results provide a clear example of how studying LLMs can make an interesting and surprising contribution to debates in cognitive science.
> The proposed benchmark tests LLM capabilities in a new domain, and there is still room for improvement so the benchmark will likely be useful for future studies as well.
> The writing is clear and there is a nice review of the relevant literature in cognitive science.

Arguments against acceptance are

- Concerns about data contamination (and task contamination) (Reviewer 1U8T)
- A lack of analysis of the model internals (Reviewers 1U8T, UkQT)
- The results do not seem to support the conclusions (Reviewers 1U8T, UkQT)
- Concerns with the experimental protocol (Reviewer 5Vv1)

While the authors argue that 3 of the reviewers should be disregarded due to `minimal or—in our view—irrelevant objections`, the AC takes these objections quite serious and finds the rebuttal insufficient in answering them (details below).

In sum, major concerns remain and the paper cannot be published at this time.

**Reviewer Concerns:**

- Concerns with the experimental protocol (Reviewer 5Vv1)

The rebuttal provides a good response, and the point is resolved.

- Concerns about data contamination (and task contamination) (Reviewer 1U8T)

The rebuttal falls short in the response (details below)

- A lack of analysis of the model internals (Reviewers 1U8T, UkQT)

The rebuttal promises results in the future, but does not add any empirical results (details below).

- The results do not seem to support the conclusions (Reviewers 1U8T, UkQT)

The rebuttal points out that some concurrent work is concurrent (and not prior), but does not properly address the actual point raised by the reviewer.

**Reviewer Scores:**

It is unlikely that Reviewer UkQT would have changed their assessment. The reviewer pointed out that the analysis might be insufficient:

> The claim would be more convincing if supported by evidence from representation-level analysis rather than behavioral absence

The reviewer provides two concurrent works that indicate that such an analysis would yield different results. Instead of answering the concerns on the merits the rebuttal instead just points out that the two papers were not prior work (which the reviewer might not have suggested it was).


Reviewer 5Vv1 might have raised their score. 3 out of 4 concerns were addressed by the rebuttal, but difference in protocol was not.


It is unlikely that Reviewer 1U8T would have changed their assessment.

> The claim and conclusion are not well supported. Since this problem type is not new, ...

The rebuttal points out that `only 12 out of 60 instruction sets appeared in the 1989 paper` and that some the 48 new examples are `rather idiosyncratic`. However the internet is vast, and the task has been around for a while. The AC is not fully convinced by the rebuttal on this point. As the authors mentioned, the original paper was quite well known (400 citations), and it is possible that more than the original 12 tasks might have leaked.

> The conclusion from the discussion is not supported by empirical evidence. There is no analysis of model internals ...

This is similar to comments by reviewer UkQT. The authors did mention that they are looking into this, but no empirical answer was provided in the rebuttal.

Reviewer Gt5a was positive and would likely have remained positive.

---

### Decision · Program_Chairs · 2026-01-26

Reject